# Methamphetamine use disorder, perceived impacts, and associated factors among adults receiving care at Sri Lanka's National Institute of Mental Health: An analytical cross-sectional study

**N. A. A. I. Nishshanka[1]⊕, T. N. L. Samarathunga[1‡], S. W. Inoka[1‡], R. Suharna[1‡], Dewarahandhi Kavishka Madushan De Silva[2]⊕, Kumarasinghe Arachchigey Sriyani🆔[1]*⊕**

**1** Department of Nursing, Faculty of Health Sciences, The Open University of Sri Lanka, Nugegoda, Colombo, Sri Lanka, **2** Department of Clinical Nursing, Faculty of Nursing, University of Colombo, Thalapathpitiya, Nugegoda, Colombo, Sri Lanka

⊕ These authors contributed equally to this work.
‡ TNLS, SWI, and RS also contributed equally to this work.
* kasri@ou.ac.lk

## Abstract

Methamphetamine addiction poses a growing public health challenge in Sri Lanka, yet limited research explores its impacts on the addicted population. This study aimed to assess the severity, patterns, and perceived impacts of methamphetamine addiction among adult patients at the National Institute of Mental Health (NIMH), Angoda, Sri Lanka. An analytical cross-sectional study was conducted among adult clients (aged >18 years) diagnosed with methamphetamine use disorder according to DSM-5 criteria at NIMH, Sri Lanka. A sample of 427 participants was recruited through purposive sampling. Data were collected using a structured, self-developed, validated, interviewer-administered questionnaire covering sociodemographic details, addiction severity (DSM-5 criteria), consumption patterns, impacts, and reasons for use. Descriptive statistics were analyzed using SPSS version 26. All participants (100%) responded to the survey. Among participants, 93.7% were male, and 65.3% were aged 18–30 years. The majority resided in urban (57.9%) or semi-urban (36.1%) areas. Addiction severity was categorized as mild (29%), moderate (38.6%), and severe (32.3%). Most (65.3%) initiated methamphetamine use between 21–30 years. Smoking (52.7%) and snorting (44.9%) were common methods of use, with peer pressure (48.9%) cited as the primary reason for initiation. The most cited physical impacts were weight loss (38.8%) and loss of appetite (37.2%), while irritability (28.8%) and interpersonal relationship problems (50.8%) were cited as common mental and social perceived impacts, respectively. Findings reveal that young urban males are predominantly affected by methamphetamine addiction, with moderate to severe dependence common. methamphetamine addiction severity was associated

**Data availability statement:** All data files are available from the figshare database https://doi.org/10.6084/m9.figshare.29816636.

**Funding:** The author(s) received no specific funding for this work.

**Competing interests:** The authors have declared that no competing interests exist.

with living arrangement, monthly income, living area, age of onset, frequency of consumption, method of consumption, and accessibility ($p < 0.05$). Peer influence and easy accessibility were significant contributing factors. The physical, mental, and social health impacts emphasize the urgent need for comprehensive intervention strategies focusing on prevention, early detection, and integrated rehabilitation services at the national level.

## Introduction

Methamphetamine is a potent central nervous system stimulant known for its strong addictive potential, which poses a critical challenge to global public health, particularly in low- and middle-income countries, where access to treatment and harm-reduction services remains limited [1,2]. The global count of amphetamine users has surged past 27 million in 2020 [3]. According to the 2021 National Survey on Drug Use and Health (NSDUH), More than 16.8 million people aged 12 or older used methamphetamine at least once during their lifetime [4].

This global trend is reflected in Asia, where methamphetamine use has significantly increased in recent years. Thailand and the Philippines were reported as the highest-rated methamphetamine prevalence in Asia [4]. In Sri Lanka, the drug abuse landscape is shifting, with methamphetamine becoming a prevalent concern [5]. In 2022, Methamphetamine was ranked as the third most commonly abused substance in Sri Lanka [6]. That year, 22,631 individuals were detained for methamphetamine-related offenses, and approximately 241 kg of the drug was seized [7]. By 2023, the number of arrests had risen to 26,096, an increase of roughly 15.3% highlighting the intensifying nature of the problem [8]. This report also underscored the gender disparity, with an overwhelmingly male representation, as 99.65% of individuals seeking treatment identified as male [7].

The reasons for methamphetamine use are complex and multifactorial. For most users, it serves as a performance enhancer, temporarily boosting energy, alertness, and concentration, fostering its use among adults engaged in high-pressure environments [9]. Moreover, social factors, including peer influences, cultural acceptance, and the accessibility of the drug, contribute to its appeal [6]. Several users also reported that past experiences of trauma, mental health issues, and familial drug use histories elevate the likelihood of methamphetamine use [10]. However, the motivations behind methamphetamine use were not clear in the Sri Lankan context.

Methamphetamine addiction affects individuals across physical, psychological, and social domains [11]. Physiologically, chronic use leads to significant cardiovascular problems, including increased heart rate and blood pressure, which can culminate in cardiovascular collapse [12]. Other than that, users also reported weight loss, dental issues colloquially known as "meth mouth," and skin infections due to increased scratching and neglect of personal hygiene [13]. Psychologically, users reported a range of mental health issues including immediate euphoria coupled with anxiety,

depression, and psychosis [14]. Cognitive impairments, including poor decision-making and impulse control, are often associated with changes in prefrontal cortex function [15]. Moreover, the cycle of withdrawal enhances the psychological burden; users experience a range of withdrawal symptoms, including dysphoria, insomnia, and intense drug cravings, which can perpetuate the cycle of addiction [16].

Moreover, methamphetamine addiction devastates interpersonal relationships and disrupts family dynamics, leading to heightened social stigma against users and their families [17]. Compulsive behaviors due to methamphetamine addiction can lead users to engage in criminal activities, high-risk sexual behaviors, which complicates the public health efforts [17,18].

Despite the rising prevalence of methamphetamine use in Sri Lanka and its well-documented socio-health impacts globally, there is a critical lack of detailed, community-based research exploring the local dynamics of use. Specifically, existing data are largely drawn from arrest records and generalized treatment statistics, which fail to capture the nuanced motivations, behavioral patterns, and perceived health consequences experienced by individuals using methamphetamine. Without such context-specific evidence, it is difficult to develop targeted interventions, policy responses, or harm-reduction strategies that are culturally and socially appropriate. In light of this gap, the current study seeks to generate in-depth, empirical insights into methamphetamine addiction among adults in Sri Lanka.

The specific objectives are to:

- assess the severity and patterns of methamphetamine use;

- determine the prevalence and types of polydrug use among methamphetamine users;

- explore self-reported motivations for methamphetamine use;

- examine users' perceptions of the perceived physical, psychological, and social impacts of their drug use; and

- identify factors associated with methamphetamine use

## Materials and methods

### Study design and setting

An analytical cross-sectional study was designed to assess methamphetamine addiction and its perceived impact on adult patients who were admitted to the National Institute of Mental Health (NIMH) at Mulleriyawa, Sri Lanka. NIMH is the country's leading tertiary-level psychiatric care facility, providing inpatient, outpatient, and community-based services and serving as a referral center for substance use and mental health disorders. The study setting was selected due to its central role in addiction treatment nationally and its access to a diverse clinical population.

### Population

The study's target population consisted of all adult individuals (more than 18 years old) receiving treatment for methamphetamine addiction in the wards and day center at the NIMH during the study period. The decision to include only adults was based on several considerations. Firstly, the clinical presentation, patterns of substance use, psychosocial consequences, and treatment modalities for methamphetamine addiction differ markedly between adolescents and adults; thus, including only adults ensured a more homogeneous sample and increased the internal validity of the findings. Further, the focus on adult patients aligns with the core patient population of the NIMH's addiction treatment services, enhancing the study's relevance to national policy and clinical practice.

Both male and female adult individuals who were diagnosed with methamphetamine substance abuse disorder according to DSM-5 criteria (F15.10, F15.15, and F15.20) as confirmed by consultant psychiatrists or psychiatric medical officers using structured clinical interviews and medical records and were currently receiving treatments were included in the

current study, while individuals with cognitive impairment, withdrawal symptoms and acute intoxication of methamphetamine were excluded from the study. Withdrawal symptoms were identified through medical records and clinical observation by attending clinicians prior to recruitment.

## Sampling and sample size

The sample size for the study was determined using Daniel's sample size calculation formula. Assuming a 95% confidence level (Z = 1.96), a prevalence (P) of 50%, and a precision (d) of 0.05, the calculated sample size was 384 [19]. To account for a potential 10% non-response rate (d), the sample size was adjusted using the formula N = n/1 − d, resulting in a final required sample size of 427 participants. The list of eligible patients was identified with the support of clinical staff using ward and day center admission logs and verified through medical records. Clinical staff initially screened patients based on diagnosis and treatment status. Participants were then recruited using a purposive sampling technique based on the predefined inclusion and exclusion criteria because the population represents a specific subgroup within the broader psychiatric patient population.

## Data collection tool

A structured, interviewer-administered questionnaire was utilized to collect the data. It was developed by referring to existing literature related to substance abuse and methamphetamine addiction [20–24]. It was comprised of four sections. The first section of the questionnaire was dedicated to obtaining participants' socio-demographic data, which had ten items including age, gender, ethnicity, civil status, family status, educational level, occupation, monthly income, residence area, and province of living. The second section assessed the severity of addiction according to routinely used DSM-5 criteria confirmed by consultant psychiatrists or psychiatric medical officers [25]. It consisted of 11 items on symptoms of addiction, and the level of severity was categorized as mild, moderate, and severe according to the number of symptoms present with the addicted individual (mild = 2 − 3, moderate = 4 − 5, severe = 6 − 11) [25]. The third section of the questionnaire included self-developed, nine items to determine the pattern of methamphetamine consumption, including age of initiation, method of use, frequency of use, accessibility to methamphetamine, last use, person who introduced methamphetamine, daily expenditure to use methamphetamine, and use of other substances (polydrug use). In addition, three open-ended items were used to collect information regarding the physical, psychological, and social impact of methamphetamine use. The last section of the questionnaire included self-reported reasons (10 items with 'yes' or 'no' responses) for methamphetamine consumption.

The content validity of the questionnaire was ensured with experts' opinions, including a consultant psychiatrist, a nursing academic, and a trained psychiatric nurse, and necessary modifications were made. The questionnaire was pre-tested among ten patients who were being treated for methamphetamine addiction to improve its clarity and examine whether the respondents could understand the items they were expected to answer [26]. The questionnaire was finalized considering the participants' opinions received during the pre-test. Specifically, participants in the pre-test provided feedback regarding the clarity and phrasing of certain items. Based on this input, we made minor revisions to simplify language and ensure better comprehension, particularly for items related to the frequency of use and perceived psychological effects. Patients who participated in the pre-test were not included in the main study.

## Data collection

Data collection was commenced after obtaining ethical clearance for the study from the Ethics Review Committee of NIMH and permission from the relevant hospital authorities. Data were collected from 20th July 2023 to 20th October 2023. All selected patients were fully informed about the purpose, nature, risks, and benefits of the study verbally and through an information sheet and obtained written consent before their participation. Volunteer participation was encouraged.

Data were collected by four investigators who had undergone a training session on data collection using an interviewer-administered questionnaire. It was done with strict adherence to patient privacy and confidentiality, ensuring that data collection did not interfere with the patient's treatment or care within their ward setup or daycare center. Interrater reliability was assessed using Cohen's Kappa, which yielded a value of 0.78 for the severity items and 0.85 for the self-reported reasons, indicating substantial agreement between the two raters. To ensure interrater reliability of perceived impact, two researchers independently coded the transcripts and then met to discuss discrepancies until consensus was reached, ensuring consistent application of codes.

## Ethical considerations

Ethical approval for the study was obtained from the Ethics Review Committee of NIMH (ERC No: 205/03/2023). Permission to access the setting and participants was obtained from the Director of the NIMH and Consultant Psychiatrists. Although a formal capacity assessment was not conducted, only clinically stable participants, without cognitive impairment, and not in a state of acute intoxication or withdrawal, were approached for consent. Clinical staff ensured that participants were coherent, alert, and oriented during recruitment. Participants were eligible only if they; were alert, fully oriented to person, place, and time; demonstrated the ability to understand the study purpose and procedures; were able to paraphrase the information sheet in their own words; could weigh risks and benefits and express a voluntary decision without coercion. All interviews were conducted in Sinhala, or Tamil, depending on participant preference. Because some items included psychological symptoms and possible suicidal ideation, a distress management protocol was followed. If a participant exhibited signs of distress, disclosed suicidal thoughts, or requested support: the interview was paused immediately; the participant was referred to the on-site psychiatrist or duty medical officer; participation continued only if the clinician confirmed the participant could safely do so. No adverse events were reported during data collection.

All individuals were provided with detailed information about the study and gave written informed consent before participation. The consent procedure, including the criteria for participant inclusion, was reviewed and approved by the aforementioned Ethics Review Committee. Permission to withdraw from the study was granted to the participants, and anonymity and confidentiality of the participants were ensured.

## Data analysis

The collected data were coded and entered into SPSS version 26 for the analysis. No missing data were reported in the dataset. The dataset was cleansed to detect any missing values or outliers. Data were descriptively analyzed for frequencies and percentages. Methamphetamine addiction was categorized into mild, moderate, and severe based on the DSM-5 established standard framework [24]. Chi-square and Fisher's exact test were utilized to derive the associated factors for methamphetamine use severity. Given the exploratory and descriptive nature of this study, no formal correction for multiple testing was applied. However, the findings are interpreted cautiously, focusing on the strength and consistency of associations rather than statistical significance alone [27]. Open-ended responses were systematically analyzed using a thematic coding framework. Initial codes were developed inductively from the data and refined through iterative reading. Two independent researchers applied the codes, and discrepancies were resolved through discussion, with a third researcher providing arbitration when necessary. Each response could receive multiple codes. Themes were then organized hierarchically into broader categories to capture patterns across responses. The final codebook with operational definitions is provided in Supplementary File S1 Table, and exemplar quotes for each major theme are presented in Supplementary File S2 Table. Percentages for perceived impacts were calculated using the total sample (N = 427) as the denominator. Because multiple responses per participant were permitted, the summed percentages exceed 100%. Intercoder reliability was assessed using Cohen's kappa for overall physical, psychological and social domains as reported as $\kappa = 0.82$.

## Results

### Sociodemographic characteristics of the participants

Out of the calculated sample size of 427, all responses were collected, yielding a 100% response rate. Table 1 presents the socio-demographic data of the participants. Of the sample, the majority were males (n = 400, 93.7%). Most of the participants were within the young adult age range of 18–30 years (n = 279, 65.3%). The findings revealed a diverse racial composition, with the Sinhala ethnic group being the most prominent, comprising 59% (n = 252) of the total participants. A total of 42.9%(n = 183) of respondents were single. A total of 59.5%(n = 254) of individuals had attained education up to the secondary level. Most of the participants (n = 292, 68.4%) resided in the Western Province, with a significant proportion from North Western Province (n = 61, 14.3%). The majority of participants resided in urban (n = 57.9%) and suburban (36.1%) areas in the country.

### Methamphetamine addiction severity and the pattern of addiction among the participants

The findings of the severity and patterns of methamphetamine addiction among participants are presented in Table 2. Addiction severity was categorized as mild (29%), moderate (38.6%), and severe (32.3%). This distribution indicates that a significant proportion of participants (over 70%) exhibit moderate to severe levels of addiction, suggesting a high treatment need within this population. The age of onset was predominantly between 21–30 years (65.3%), with smaller proportions starting at ages 12–20 (22.5%) and 31–40 (9.8%) years. Consumption frequency varied, with 48.2% using methamphetamine several days a week, 28.1% using it daily, and 22.3% using it weekly. The majority reported their last usage within a week (54.6%), while 22.7% had used it within a month. Smoking (52.7%) and snorting (44.9%) were the most common methods of consumption, with minimal use of injection (1.9%) or swallowing (0.5%). Most participants were introduced to methamphetamine by friends (83.8%), followed by relatives (11%). Accessibility to the substance was perceived as fairly easy (46.6%) or easy (36.5%) by the majority, while 5.6% found it difficult to access.

### Polydrug consumption

Out of the sample, 55 (12.9%) used methamphetamine alone while the rest of them (n = 372, 87.1%) used other drugs with methamphetamine. Among the polydrug users, multiple responses were possible: 164 (38.4%) reported consuming alcohol, 166 (38.9%) used cannabis, 128 (30%) used heroin, 131 (30.7%) used tobacco, and 111 (26%) reported using other drugs. The high prevalence of polydrug use indicates a complex pattern of substance dependence, which can intensify health risks, complicate treatment, and heighten the potential for social harm.

To explore the potential confounding effect of polydrug use, a stratified descriptive analysis was performed comparing methamphetamine-only users (n = 55; 12.9%) with polydrug users (n = 372; 87.1%). A higher proportion of methamphetamine-only users were classified as severely addicted (40%) compared to polydrug users (31.7%). Patterns of consumption also differed. Daily use was more common among methamphetamine-only users (41.8%) than among polydrug users (26.7%), while polydrug users more frequently reported using the drug several days per week.

### Self-reported reasons for methamphetamine consumption

Participants were asked to cite the most probable reasons for their methamphetamine addiction, and individuals were allowed to cite one or more reasons. Table 3 presents the self-reported reasons for methamphetamine addiction among the participants. Peer pressure was the most frequently cited reason, with 48.9% acknowledging its influence. A smaller proportion (23.1%) reported the involvement of peers in drug-related businesses as a contributing factor. Family-related reasons, such as isolation from family (11.5%), lack of family closeness (9.1%), and lack of parental support (3.2%), were less commonly reported. Only 1.8% attributed their consumption to having parents who were drug abusers, and 1.1% cited excessive punishment. Work-related factors included maintaining attention and concentration (13.6%), increasing job productivity (10.5%), and coping with a heavy workload (2.5%).

**Table 1. Sociodemographic data of the participants (N = 427).**

| Characteristics | Frequency (n) | Percentage (%) |
|---|---|---|
| **Age (Years)** | | |
| 18-30 | 279 | 65.3 |
| 31-40 | 101 | 23.7 |
| 41-50 | 34 | 8 |
| 51-60 | 13 | 3.0 |
| **Gender** | | |
| Male | 400 | 93.7 |
| Female | 27 | 6.3 |
| **Social status** | | |
| Married | 150 | 35.1 |
| Single | 183 | 42.9 |
| Widowed | 06 | 1.4 |
| Divorced | 22 | 5.1 |
| Separated | 66 | 15.5 |
| **Education[a]** | | |
| Primary education | 63 | 14.8 |
| Secondary education | 254 | 59.5 |
| Tertiary/Higher education | 65 | 15.2 |
| Vocational education | 45 | 10.5 |
| **Ethnicity** | | |
| Sinhala | 252 | 59.0 |
| Moor | 105 | 24.6 |
| Tamil | 70 | 16.4 |
| **Living with whom** | | |
| Family | 323 | 75.6 |
| Alone | 95 | 22.3 |
| Other | 09 | 2.1 |
| **Monthly income (LKR)[b]** | | |
| <20,000 | 66 | 15.5 |
| 20,000–30,000 | 69 | 16.2 |
| 30,0001–40,000 | 136 | 31.8 |
| 40,001–50,000 | 90 | 21 |
| >50,000 | 66 | 15.5 |
| **Living province** | | |
| Western province | 292 | 68.4 |
| Southern province | 29 | 6.8 |
| Sabaragamuwa province | 03 | 0.7 |
| Eastern province | 04 | 0.9 |
| Central province | 21 | 4.9 |
| North Western province | 61 | 14.3 |
| Northern province | 17 | 4 |
| **Living area** | | |
| Urban | 247 | 57.9 |
| Semi-urban | 154 | 36.1 |
| Rural | 26 | 6 |

[a]Primary education – Grade 01 to Grade 05.

Secondary education – Grade 06 to Advanced level (A/L).

*(Continued)*

**Table 1.** (Continued)

Tertiary/Higher education – Undergraduate and/or postgraduate.

Vocational education – Vocational training or technical education.

[b]LKR; Sri Lankan rupees.

Urban – Areas within municipal/urban councils with high population density and primarily commercial or residential development.

Semi-urban: Transitional areas with mixed residential and developing commercial features, located between urban and rural settings.

Rural: Areas characterized by low population density, predominantly agricultural or village-based environments.

### Perceived physical impact of methamphetamine

Self-reported physical impacts of methamphetamine addiction were illustrated in Fig 1A and 1B. Weight loss (38.8%) and loss of appetite (37.2%) were the most commonly reported physical impacts among participants. Dental problems were reported by 12.1%, while malaise and chest pain were reported by 6.3% each. Other notable impacts included cough (7.4%), dry mouth (5.1%), myalgia (5.6%), and excessive sweating (4.2%) were also mentioned. Less frequently reported issues included physical injuries, headaches, and muscle cramps (around 3.2%−3.9% each), hair loss (2.3%), and jaw clenching (1.1%). Muscle rigidity was the least reported, affecting only 0.4% of participants.

### Perceived psychological/mental impact

The perceived psychological impacts of methamphetamine addiction are illustrated in Fig 2A and 2B. Irritability was the most frequently reported psychological symptom (28.8%), followed by delusions (24.8%) and hallucinations (22.9%). Sleep problems were also common, affecting 18.7% of participants. Anxiety and fearfulness (14.5%), feeling low (11.9%), and poor concentration and attention (8.4%) were notable issues. Suicidal thoughts or self-harm (7.9%), homicidal ideas (3%), and aggression (4.6%) were less frequently mentioned. Loss of interest (6.3%) and restlessness/agitation (1.1%) were among the least reported. These findings highlight significant mental health challenges, particularly irritability, psychotic symptoms, and anxiety-related issues, associated with methamphetamine use.

### Perceived social impact of methamphetamine

The perceived social impacts of methamphetamine addiction are illustrated in Fig 3A and 3B. Interpersonal relationship problems and conflicts were the most frequently reported social impact, affecting 50.8% of participants. Financial problems were also common, reported by 32%, while stigmatization and social isolation were noted by 28.3%. Employment disruption was reported by 10%, and legal problems by 8.9%. Poor role performance affected 7.9%, and academic difficulties were the least reported, affecting only 2.5% of participants.

### Associated factors for methamphetamine use severity

Severity of methamphetamine use was significantly associated with living arrangement ($p = 0.043$), monthly income ($p < 0.001$), living area ($p < 0.001$), age of onset ($p = 0.028$), frequency of consumption ($p < 0.001$), method of consumption ($p < 0.001$), and accessibility ($p < 0.001$) (Table 4).

## Discussion

The current study is the first study, to the best of our knowledge, investigating Methamphetamine addiction and its perceived impacts on adult individuals receiving treatment in a local context. By examining various physical, psychological, and social impacts of methamphetamine addiction, the study contributes new insights into the complexities of addiction within this local context.

**Table 2. Methamphetamine addiction severity and the pattern of addiction among the participants (N=427).**

| Characteristics | Frequency (n) | Percentage (%) |
|---|---|---|
| **Severity of addiction** | | |
| Mild | 124 | 29 |
| Moderate | 165 | 38.6 |
| Severe | 138 | 32.3 |
| **Age of onset (years)** | | |
| 12-20 | 96 | 22.5 |
| 21-30 | 279 | 65.3 |
| 31-40 | 42 | 9.8 |
| 41-50 | 10 | 2.3 |
| **Frequency of consumption[a]** | | |
| Daily | 120 | 28.1 |
| Several days in week | 206 | 48.2 |
| Weekly | 95 | 22.3 |
| Once in month | 06 | 1.4 |
| **Last usage** | | |
| Within a week | 233 | 54.6 |
| Within a month | 97 | 22.7 |
| One month ago | 66 | 15.5 |
| 06 months ago | 31 | 7.2 |
| **Method of consumption** | | |
| Smoking | 225 | 52.7 |
| Swallowing (pill) | 02 | 0.5 |
| Snorting | 192 | 44.9 |
| Injection | 08 | 1.9 |
| **Introduced by** | | |
| Friends | 358 | 83.8 |
| Relatives | 47 | 11 |
| Foreigner | 10 | 2.3 |
| Family member | 12 | 2.8 |
| **Accessibility[b]** | | |
| Difficult | 24 | 5.6 |
| Fairly difficult | 48 | 11.2 |
| Fairly easy | 199 | 46.6 |
| Easy | 156 | 36.5 |

[a]Frequency of methamphetamine use refers to participants' typical pattern of consumption prior to admission, categorized as:

• Daily: use every day or almost every day

• Several days per week: use on 3–5 days per week

• Weekly: use approximately once per week

• Once per month: use about once monthly

[b]Accessibility refers to participants' perceived ease of obtaining methamphetamine during their regular period of use, categorized as:

• Easy: readily obtainable with minimal effort

• Fairly easy: obtainable with some effort but not difficult

• Fairly difficult: required moderate effort or limited availability

• Difficult: hard to obtain or inconsistent access

**Table 3. Self-reported reasons for methamphetamine addiction among the participants.**

| Reason | Yes<br>n (%) | No<br>n (%) |
|---|---|---|
| Peer pressure | 209 (48.9) | 218 (51) |
| Peers are doing business related to ICE[a] | 99 (23.2) | 328 (76.8) |
| Isolation from family[b] | 49 (11.5) | 378 (88.5) |
| Lack of parental support | 14 (3.3) | 413 (96.7) |
| Lack of family closeness[c] | 39 (9.1) | 388 (90.9) |
| Parents are drug abusers | 8 (1.9) | 419 (98.1) |
| Subjected to excessive punishment | 5 (1.2) | 422 (98.8) |
| To increase the productivity of the job | 45 (10.5) | 382 (89.5) |
| To keep attention and concentration on work | 58 (13.6) | 369 (86.4) |
| The heavy workload of the job | 11 (2.6) | 416 (97.4) |

[a]ICE – General name used for methamphetamine crystals.

[b]Behavioral or physical separation from family.

[c]Emotional dimension of absence of warmth and supportive bond.

Isolation from family: Physical or behavioural separation from family members (e.g., spending limited time at home, living away, or avoiding family interaction).

Lack of family closeness: Emotional distance or weak supportive bonds within the family (e.g., reduced warmth, trust, or connectedness).

Lack of parental support: Perceived inadequate guidance, supervision, emotional support, or involvement from parents.

Peers doing business related to methamphetamine: Friends or acquaintances involved in selling, distributing, or trafficking methamphetamine.

Excessive punishment: Experiences of harsh or punitive disciplinary practices in childhood or adolescence.

As revealed in the present study, the distribution of participants indicates a marked concentration in the Western Province, followed by North Western Province may reflect a regional pattern of methamphetamine use and availability. The majority residing in urban and suburban areas suggest that methamphetamine addiction may be more commonly identified or reported in these settings, possibly due to higher drug availability. According to the Drug Related Statistics 2019 in Sri Lanka [28], the majority of drug-related arrests were reported from the Western Province, North Western Province, and Southern Province of Sri Lanka. While these findings support the current findings, they also highlight the need for broader surveillance and the necessity of prompt preventive strategies.

The male predominance found in the present study is aligned with existing literature, denoting that men are more likely to engage in high-risk substance use behaviors, including methamphetamine use [6,29]. Various cultural norms, social influences, and biological susceptibility may have contributed to this gender imbalance. In Sri Lankan culture, children and women are generally well protected within the family context, and substance use among women is strongly stigmatized by society's behaviors [6]. This societal disapproval may contribute to the notably low prevalence of substance use among females. Consistent with previous studies [6], most of the study participants were aged between 18–30 years, and this highlights the vulnerability of young adults to use methamphetamine. Adulthood often involves significant transitions, including entering the workforce and pursuing higher education stress and increase susceptibility to substance use [30]. These findings emphasize the importance of targeted preventive measures for young populations. The diversity in ethnic groups in the study sample implies that methamphetamine addiction transcends demographic boundaries. While this distribution partly reflects the local population, it also highlights the necessity of implementing culturally competent preventive measures to protect young adults from methamphetamine addiction. Nearly two-fifths of the participants reported being single, while a considerable number of participants were separated from their spouses. This may indicate an association

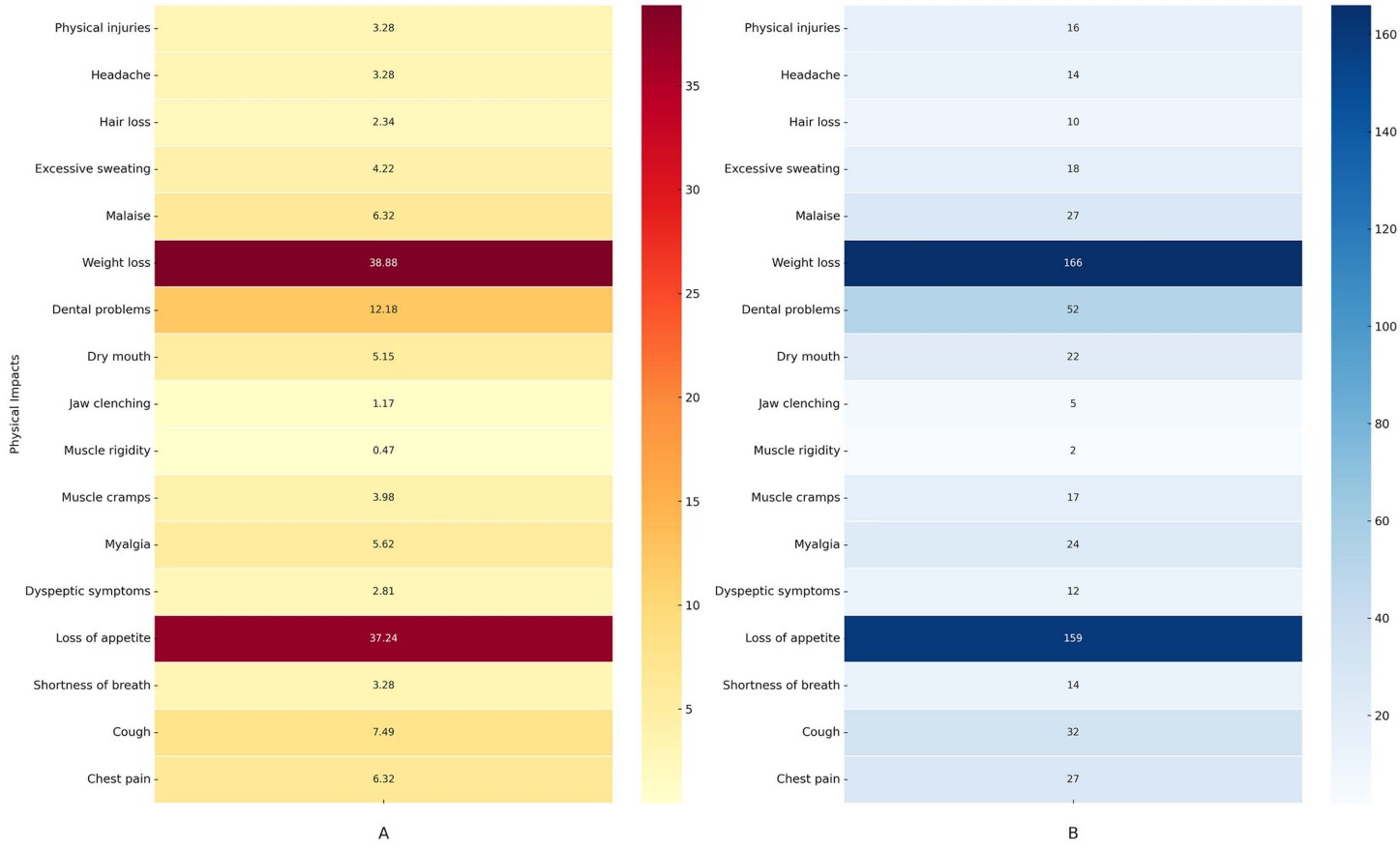

**Fig 1. Heatmaps illustrating self-reported perceived physical impacts due to methamphetamine addiction. (A)** Percentage heatmap: Illustrates the relative percentages of self-reported perceived physical impacts due to methamphetamine addiction, with color gradients representing the percentage of perceived symptoms. Darker colors indicate a higher percentage of self-reported physical impact, while lighter colors indicate a lower percentage of perceived effects. **(B)** Frequency heatmap: Illustrates the relative frequency of self-reported perceived physical impacts due to methamphetamine addiction, with color gradients representing the frequency of perceived symptoms. Darker colors indicate a higher frequency of self-reported physical impact, while lighter colors indicate a lower frequency of perceived effects.

between methamphetamine addiction and disrupted family relationships, suggesting that substance use could be linked with marital instability [31]. Compared with the findings of the National Dangerous Drug Control Board report in 2022 [32], more methamphetamine -addicted individuals in the present study were separated (66 vs 12), and this suggests a possible link between methamphetamine use and family separation.

## Addiction to methamphetamine

The findings on Methamphetamine addiction severity and patterns of use among participants reveal critical insights into the prevalence, onset, frequency, and influencing factors of methamphetamine use. As revealed in the present study, most of the participants had moderate levels of methamphetamine addiction, followed by 32.3% classified as severe and 29% as mild. This distribution suggests a significant proportion of users have progressed beyond mild addiction, highlighting the need for intensive intervention and rehabilitation programs tailored to different addiction severities. The present study findings demonstrated a high frequency of methamphetamine consumption. Nearly half reported use on several days per week, while 28.1% consumed it daily. This pattern suggests a high dependency risk, particularly among daily users.

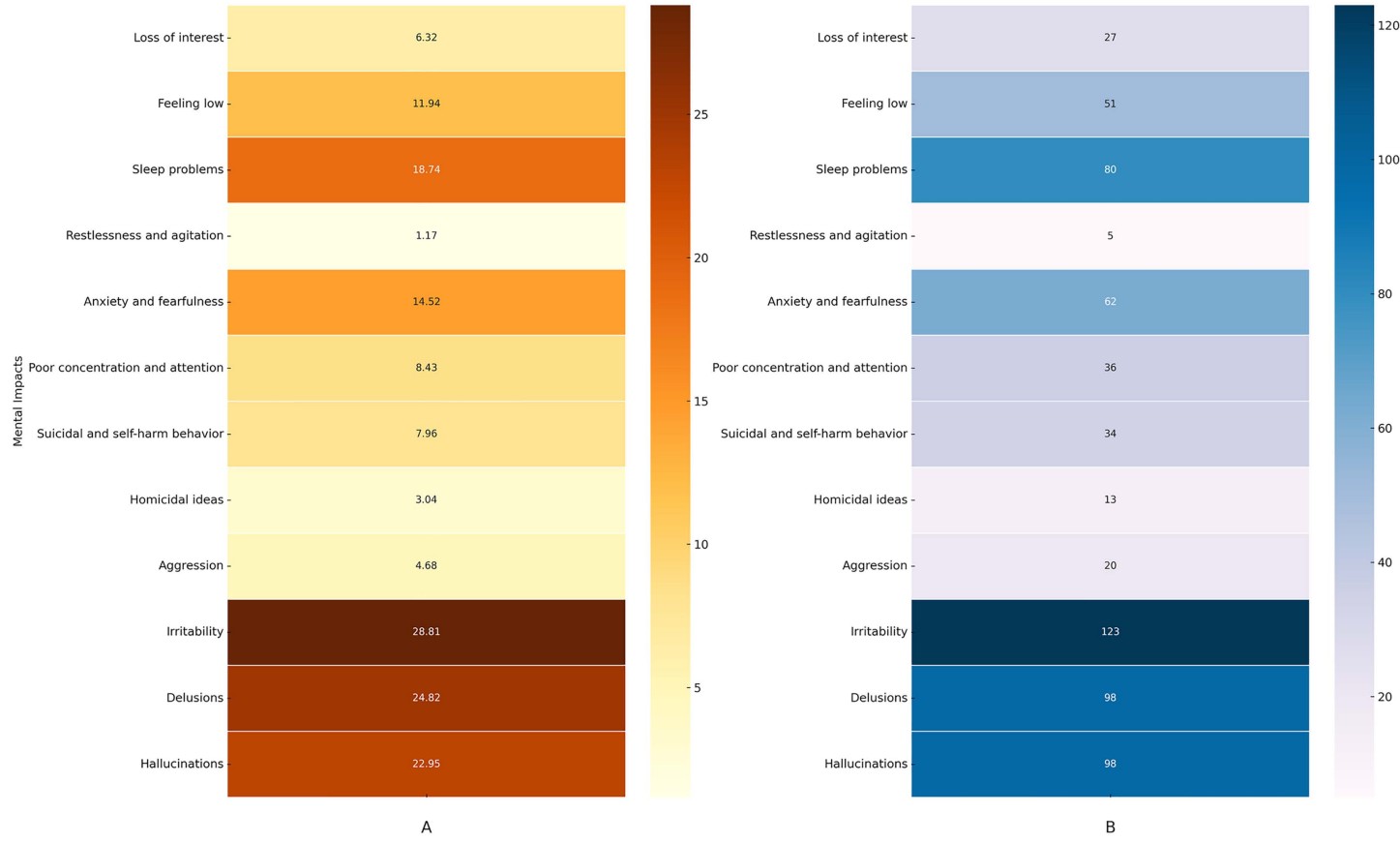

**Fig 2. Heatmaps illustrating self-reported perceived psychological/mental impacts due to methamphetamine addiction. (A)** Percentage heatmap: Illustrates the relative percentages of self-reported perceived psychological/mental impacts due to methamphetamine addiction, with color gradients representing the percentage of perceived symptoms. Darker colors indicate a higher percentage of self-reported psychological/mental impact, while lighter colors indicate a lower percentage of perceived effects. **(B)** Frequency heatmap: Illustrates the relative frequency of self-reported perceived psychological/mental impacts due to methamphetamine addiction, with color gradients representing the frequency of perceived symptoms. Darker colors indicate a higher frequency of self-reported psychological/mental impact, while lighter colors indicate a lower frequency of perceived effects.

Additionally, more than half reported using methamphetamine within the past week. Prolonged methamphetamine use has been associated with various disorders in previous studies [33]. According to the Morbidity and Mortality Weekly Report published in the USA, 50% of persons using methamphetamine in the past year met diagnostic criteria for past-year meth-amphetamine use disorder [29]. Smoking and snorting are the most common methods of methamphetamine consumption in the present study. This finding is consistent with the report by the American Addiction Centre, most consumers reported the above methods of consumption [34]. A survey conducted in Los Angeles revealed that the most popular method of methamphetamine use among participants were smoking (74.8%), followed by snorting (65.4%) [35]. These routes of administration have been widely reported in the literature to be associated with more rapid drug absorption and a higher potential for dependence [36].

As revealed in the present study, the majority of participants were introduced to methamphetamine by friends, followed by relatives. The accessibility of methamphetamine is a considerable concern in the present study. As found a significant proportion of participants reported that methamphetamine was fairly easy to access, while more than one-third found it easily accessible. In contrast, only a small percentage considered it difficult to obtain. This issue was similarly noted in Los

                                    

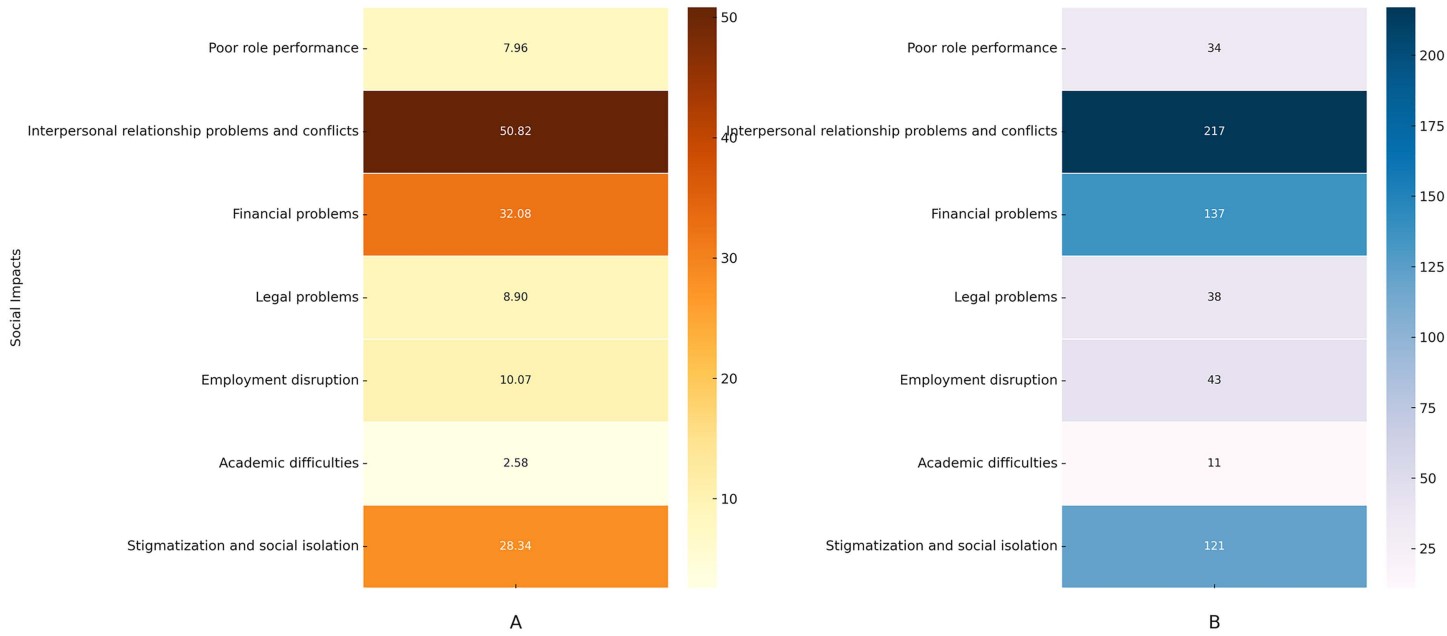

**Fig 3. Heatmaps illustrating self-reported perceived social impacts due to methamphetamine addiction.** **(A)** Percentage heatmap: Illustrates the relative percentages of self-reported perceived social impacts due to methamphetamine addiction, with color gradients representing the percentage of perceived symptoms. Darker colors indicate a higher percentage of self-reported social impact, while lighter colors indicate a lower percentage of perceived effects. **(B)** Frequency heatmap: Illustrates the relative frequency of self-reported perceived social impacts due to methamphetamine addiction, with color gradients representing the frequency of perceived symptoms. Darker colors indicate a higher frequency of self-reported social impact, while lighter colors indicate a lower frequency of perceived effects.

Angeles, where 41.1% of their participants perceived that it was easy to access methamphetamine around their neighborhood [35]. The perception of easy access to methamphetamine among treatment-seeking adults at NIMH is concerning as it may contribute to continued use or relapse within this group e [37]. While these observations cannot be generalized nationally, they highlight the importance of supporting patients within the communities where they live through targeted-context specific interventions that address the environmental factors reported by this population.

## Self-reported reasons for methamphetamine addiction

The findings on self-reported reasons for methamphetamine addiction indicate that peer influence is the most significant factor, with nearly half of the participants citing peer pressure as the reason for their drug use. The current findings align with a South African study, which mentioned the same results, that friends were by far the most common people to introduce methamphetamine to the participants [38]. This concern highlights the strong role of peer influence in substance addiction, emphasizing the need for peer-focused prevention programs that address risky behaviors in social circles [39]. This highlights the strong role of social circles in substance initiation and suggests that prevention strategies should focus on peer-led education and awareness programs. Approximately one in four participants reported that their peers were involved in methamphetamine-related business, which could indicate exposure to drug distribution networks that encourage or normalize methamphetamine use. Family-related factors such as isolation from family, lack of family closeness, and lack of parental support were reported by a smaller proportion of participants. In agreement with present findings, a review study identified coping with family-related problems and psychological distress as common reasons for methamphetamine use [40]. Similarly, a small proportion of participants reported having parents who were drug users, and experiencing excessive punishment during childhood as reasons for their addiction. These findings suggest that although family

**Table 4. Associated factors for methamphetamine use severity.**

| Characteristics | Mild | Moderate | Severe | p-value |
|---|---|---|---|---|
| **Age (Years)** | | | | |
| 18-30 | 77 (27.6) | 110 (39.4) | 92 (33) | 0.232* |
| 31-40 | 26 (25.7) | 42 (41.6) | 33 (32.7) | |
| 41-50 | 15 (44.1) | 08 (23.5) | 11 (32.4) | |
| 51-60 | 06 (46.2) | 05 (38.5) | 02 (15.4) | |
| **Gender** | | | | |
| Male | 117 (29.3) | 152 (38) | 131 (32.8) | 0.567 |
| Female | 07 (25.9) | 13 (35.6) | 07 (25.9) | |
| **Social status** | | | | |
| Married | 38 (25.3) | 60 (40) | 52 (34.7) | 0.374* |
| Single | 55 (30.1) | 72 (39.3) | 56 (30.6) | |
| Widowed | 01 (16.7) | 02 (33.3) | 03 (50) | |
| Divorced | 12 (54.5) | 05 (22.7) | 05 (22.7) | |
| Separated | 18 (27.3) | 26 (39.4) | 22 (33.3) | |
| **Education[a]** | | | | |
| Primary education | 25 (39.7) | 23 (36.5) | 15 (23.8) | 0.187 |
| Secondary education | 65 (25.6) | 107 (42.1) | 82 (32.3) | |
| Tertiary/Higher education | 22 (33.8) | 20 (30.8) | 23 (35.4) | |
| Vocational education | 12 (26.7) | 15 (33.3) | 18 (40) | |
| **Ethnicity** | | | | |
| Sinhala | 68 (27) | 94 (37.3) | 90 (35.7) | 0.369 |
| Moor | 31 (29.5) | 45 (42.9) | 29 (27.6) | |
| Tamil | 25 (35.7) | 26 (37.1) | 19 (27.1) | |
| **Living with whom** | | | | |
| Family | 93 (28.8) | 119 (36.8) | 111 (34.4) | 0.043* |
| Alone | 31 (32.6) | 41 (43.2) | 23 (24.2) | |
| other | 00 (0) | 05 (55.6) | 04 (44.4) | |
| **Monthly income (LKR)[b]** | | | | |
| <20,000 | 37 (56.1) | 23 (34.8) | 06 (9.1) | <0.001 |
| 20,000–30,000 | 18 (26.1) | 34 (49.3) | 17 (24.6) | |
| 30,0001–40,000 | 54 (39.7) | 44 (32.4) | 38 (27.9) | |
| 40,001–50,000 | 12 (13.3) | 39 (43.3) | 39 (43.3) | |
| >50,000 | 03 (4.5) | 25 (37.9) | 38 (57.6) | |
| **Living area** | | | | |
| Urban | 53 (21.5) | 95 (38.5) | 99 (40.1) | <0.001* |
| Semi-urban | 54 (35.1) | 66 (42.9) | 34 (22.1) | |
| Rural | 17 (65.4) | 04 (15.4) | 05 (19.2) | |
| **Age of onset (years)** | | | | |
| 12-20 | 38 (39.6) | 39 (40.6) | 19 (19.8) | 0.028* |
| 21-30 | 74 (26.5) | 106 (38) | 99 (35.5) | |
| 31-40 | 08 (19) | 18 (42.9) | 16 (38.1) | |
| 41-50 | 04 (40) | 02 (20) | 04 (40) | |
| **Frequency of consumption** | | | | |
| Daily | 35 (29.2) | 30 (25) | 55 (45.8) | <0.001* |
| Several days in week | 55 (26.7) | 85 (41.3) | 66 (32) | |

*(Continued)*

**Table 4.** (Continued)

| Characteristics | Mild | Moderate | Severe | p-value |
|---|---|---|---|---|
| Weekly | 34 (35.8) | 48 (50.5) | 13 (13.7) | |
| Once in month | 00 (0) | 02 (33.3) | 04 (66.7) | |
| **Method of consumption** | | | | |
| Smoking | 50 (22.2) | 95 (42.2) | 80 (35.6) | <0.001* |
| Swallowing (pill) | 00 (0) | 00 (0) | 02 (100) | |
| Snorting | 74 (38.5) | 68 (35.4) | (26) | |
| Injection | 00 (0) | 02 (25) | (75) | |
| **Accessibility** | | | | |
| Difficult | 16 (66.7) | 04 (16.7) | 04 (16.7) | <0.001* |
| Fairly difficult | 18 (37.5) | 14 (29.2) | 16 (33.3) | |
| Fairly easy | 60 (30.2) | 76 (38.2) | 63 (31.7) | |
| Easy | 30 (19.2) | 71 (45.5) | 55 (35.3) | |

*Fisher's exact test.

dynamics may influence drug use, they are not the primary drivers of methamphetamine addiction in the study population [41]. Work-related reasons for methamphetamine use were reported by a small but notable percentage of participants. Approximately 13% used methamphetamine to maintain attention and concentration at work, while 10.5% reported using it to enhance job productivity. The association between substance use and work performance indicates the need for work-place mental health programs and stress management interventions to reduce reliance on drugs for job-related demands.

## Polydrug use

Present findings indicate that while a small portion of participants exclusively use methamphetamine, the vast majority engage in polydrug use. These findings resonate with the findings of a previous study where polydrug consumption was very common among methamphetamine consumers [42]. This high rate of concurrent substance use underscores the complex nature of addiction, suggesting that individuals may use multiple drugs to achieve specific effects or to self-medicate varying psychological or physical needs. Within this polydrug subgroup, notable substances commonly used alongside methamphetamine include alcohol, cannabis, heroin, and tobacco, with a considerable percentage also reporting the use of other unspecified drugs. As revealed in a case series study in the local context, young adults consumed a mixture of substances, including alcohol, heroin, cannabis, and amphetamines [5]. As found in previous studies, polydrug use was associated with rule-breaking behavior [30], violent and traumatic behavior, physical and sexual abuse [5], and various other psychiatric disorders such as major depression, post-traumatic stress disorder, panic attacks, obsessive-compulsive disorder, and antisocial personality [43]. As cited by earlier [5], compared with mature adult brains, young adults are more vulnerable to drug-seeking behaviors due to the "pleasure-seeking behavior" of the young, immature brains [44]. The presence of polydrug use highlights the urgent need for integrated and context-specific treatment models that address the complexity of multiple substance dependencies. Effective withdrawal management requires improved clinical protocols and trained personnel to manage compound risks. There is a critical need for culturally adapted relapse prevention strategies and community-based follow-up services to support sustained recovery, especially in low-resource settings [45].

The very high prevalence of polydrug use in this sample (87%) presents important interpretive challenges when examining the impacts of methamphetamine. To address this, the present study conducted stratified analyses comparing methamphetamine -only users with polydrug users. Notably, methamphetamine -only users showed a higher proportion of severe addiction and a greater likelihood of daily use. These findings suggest that, within this clinical population, heavy or

frequent methamphetamine use can occur independently of other substances. Many of the "perceived impacts," such as aggression, irritability, interpersonal relationship problems, and loss of appetite, could be potentiated or confounded by the simultaneous use of depressants like alcohol and stimulants such as cannabis or tobacco. Similar findings were reported by Darke et al. [24] and Jayanthi et al. [46], who noted that in polydrug contexts, methamphetamine 's direct neurotoxic and behavioral effects are often amplified or masked by other substances. Therefore, rather than attributing adverse outcomes solely to methamphetamine, it is more accurate to view them as the result of interactive or synergistic drug effects. This emphasizes the importance of future research employing stratified analyses or biological verification methods to isolate methamphetamine -specific consequences while accounting for concurrent substance use.

## Perceived physiological impact of methamphetamine addiction

The findings highlight a range of self-reported physical impacts of methamphetamine use, demonstrating its potential to affect multiple bodily systems. The most frequently reported effects, including weight loss and loss of appetite, were commonly observed among users and may indicate a risk of nutritional deficiencies over time. These findings underscore the need for targeted nutritional interventions to address malnourishment and to restore metabolic balance in affected individuals [47]. Similarly, comprehensive dental care is essential as methamphetamine use has been commonly associated with notable oral health concerns [48]. Other symptoms, such as malaise, chest pain, cough, and excessive sweating, further demonstrate the extent of the influence of methamphetamine addiction on the human body.. As reported by Darke [23] methamphetamine addiction has been associated with physical harm, including toxicity, cardiovascular/cerebrovascular pathology, dependence, and blood-borne virus transmission.

## Perceived psychological impacts

Previous studies have documented the psychological harm of methamphetamine addiction, including psychosis, depression, suicide, anxiety, and violent behaviors [23]. The present findings highlight range and severity of perceived psychological effects of methamphetamine use, offering insights into both short-term disturbances and long-term mental health challenges. Irritability was the most prevalent symptom, indicating increased emotional instability, which may be negatively affects interpersonal relationships and potentially lead to aggression or conflict [49–51]. In addition, the high prevalence of delusions and hallucinations suggests a link between methamphetamine use and psychotic symptoms, which can emerge or intensify with frequent or prolonged use. Similarly, many studies have reported hallucinations as a frequently found psychotic symptom of chronic methamphetamine use [51,52]. Compared with previous studies, sleep problems were reported to a considerable extent in the present study, suggesting a connection between methamphetamine use and disturbed sleep-wake cycles [44,53]. These disturbances may contribute to fatigue, impaired judgment, and worsening mood swings [44]. Anxiety is more common among chronic methamphetamine users [44,51,54,55]. High levels of anxiety/fearfulness and feeling low found in the present study highlight the drug's ability to influence the emotional and psychological well-being of users. These issues, tied with poor concentration and attention, can have profound implications for daily functioning and quality of life, often interfering with work and education [56]. Of particular concern, suicidal thoughts or self-harm behaviors were reported by 7.9% of participants [52]. Although homicidal ideas, aggression, and loss of interest were comparatively lower, these rates still signify a substantial risk that emphasizes the need for comprehensive mental health screenings and providing necessary treatments.

## Perceived social impact of methamphetamine addiction

The most prominent social impact reported was interpersonal relationship problems and conflict affecting, over half of the participants. This finding underscores that methamphetamine use may exert on personal connections, contributing to family disputes, breakdowns in friendships, and overall social dysfunction. Financial problems were another commonly

reported challenge, suggesting a potential association between methamphetamine use and economic strain. Additionally, stigmatization and social isolation reflect the broader social barriers and negative attitudes that individuals with substance use disorders often face, which can hinder access to support and worsen mental health outcomes. Although employment disruption (10%, legal problems (8.9%), poor performance, and academic difficulties were less frequently reported, they still indicate that methamphetamine addiction may affect nearly every aspect of daily life, from maintaining steady employment to fulfilling societal and personal responsibilities. As reported previously, chronic use of methamphetamine causes social isolation due to social withdrawal [21].

### Factors associated with methamphetamine use

In the present study, several socio-demographic and contextual factors were significantly associated with the severity of methamphetamine use, including monthly income, living area, age of onset, frequency and method of consumption, accessibility, and living arrangements. Notably, individuals with higher income and those residing in urban areas showed a greater likelihood of severe use, which contrasts with studies from the United States and other settings where low socio-economic status and rural residence were more strongly linked to methamphetamine use and related harms [57,58]. This difference may reflect local factors, including greater drug availability and stronger enforcement activity in urban areas as well as the NIMH catchment pattern, affordability of the drug among higher-income users, and potential referral or reporting variations. These contextual influences may shape the associations observed among treatment-seeking adults at NIMH [59]. Earlier onset of use (12–20 years) in the present study sample was associated with milder severity, diverging from global evidence that early initiation is associated with more severe dependency and poorer outcomes [57]. This discrepancy may indicate cohort or recall differences, or that early initiators in this sample have not yet escalated to severe use. In contrast, present findings on frequency and method of consumption aligned with previous research, showing that daily use, smoking, and especially injection were strongly associated with severe dependence [57,58] Furthermore, reports of easy access to methamphetamine among participants coincided with more severe use patterns, which aligns with previous observations that drug availability may be linked to both initiation and escalation in other contexts [60]. Overall, while some associations replicate international findings, others highlight unique local dynamics that underscore the importance of context-specific interventions for prevention and harm reduction.

### Strengths and limitations

The present study has several strengths, including a 100% response rate and comprehensive coverage of the impact of methamphetamine. However, it also has certain limitations. The descriptive cross-sectional design restricts causal inference, and findings represent associations rather than cause–and–effect relationships. Reliance on self-reported data may introduce recall social desirability bias, such as under- or over-reporting. Therefore, it should be interpreted with caution. Additionally, because all participants were recruited from a single tertiary mental health institution, the results may not fully reflect community users or those untreated for methamphetamine use disorder. The study also did not control for potential confounding from polydrug use or comorbid psychiatric conditions, which may have influenced reported impacts.

Future studies should employ longitudinal and multicenter designs, include biochemical verification of substance use, and explore qualitative dimensions to better understand user experiences.

### Conclusions

This study provides empirical evidence on the growing public health concern of methamphetamine addiction in Sri Lanka, particularly among young adult males. It identifies the high prevalence of moderate to severe addiction, influenced predominantly by peer pressure and facilitated by easy access to methamphetamine. Often accompanied by polydrug use and the physical, psychological, and social repercussions, including disrupted family relationships and workplace

challenges, underscore the multifaceted impact of methamphetamine use. The severity of methamphetamine use is associated with living arrangements, monthly income, living area, age of onset, frequency of consumption, method of consumption, and accessibility. The study bridges a significant research gap by providing localized insights into patterns and consequences of methamphetamine addiction. These findings highlight an urgent need for targeted, youth-focused prevention strategies, particularly peer-led and culturally tailored interventions. Additionally, tighter regulation and enforcement to disrupt methamphetamine supply chains are essential to curb availability and mitigate further harm.

## Supporting information

**S1 Table. 1, 2, 3 codebook.**
(DOCX)

**S2 Table. 1, 2, 3 exemplar quotes.**
(DOCX)

## Acknowledgments

The authors would like to thank all the participants and the NIMH clinical staff and administration officers.

## Author contributions

**Conceptualization:** N. A. A. I. Nishshanka, T. N. L. Samarathunga, S. W. Inoka, R. Suharna, Kumarasinghe Arachchigey Sriyani.

**Data curation:** N. A. A. I. Nishshanka, Dewarahandhi Kavishka Madushan De Silva.

**Formal analysis:** N. A. A. I. Nishshanka, Dewarahandhi Kavishka Madushan De Silva.

**Investigation:** N. A. A. I. Nishshanka, T. N. L. Samarathunga, S. W. Inoka, R. Suharna, Kumarasinghe Arachchigey Sriyani.

**Methodology:** N. A. A. I. Nishshanka, T. N. L. Samarathunga, S. W. Inoka, R. Suharna, Dewarahandhi Kavishka Madushan De Silva, Kumarasinghe Arachchigey Sriyani.

**Project administration:** Kumarasinghe Arachchigey Sriyani.

**Supervision:** Kumarasinghe Arachchigey Sriyani.

**Writing – original draft:** Dewarahandhi Kavishka Madushan De Silva, Kumarasinghe Arachchigey Sriyani.

**Writing – review & editing:** Dewarahandhi Kavishka Madushan De Silva, Kumarasinghe Arachchigey Sriyani.

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
