## [Decision Letter · Decision Letter 0]

8 Jul 2025

Dear Dr. Sriyani,

Thank you for submitting your manuscript to PLOS ONE. After careful consideration, we feel that it has merit but does not fully meet PLOS ONE’s publication criteria as it currently stands. Therefore, we invite you to submit a revised version of the manuscript that addresses the points raised during the review process.

We look forward to receiving your revised manuscript.

Kind regards,

Nicholas Aderinto Oluwaseyi

Academic Editor

PLOS ONE

Journal Requirements:

2. Please describe in your methods section how capacity to provide consent was determined for the participants in this study. Please also state whether your ethics committee or IRB approved this consent procedure. If you did not assess capacity to consent please briefly outline why this was not necessary in this case.

Reviewers' comments:

Reviewer's Responses to Questions

**Comments to the Author**

1. Is the manuscript technically sound, and do the data support the conclusions?

Reviewer #1: No

Reviewer #2: Yes

2. Has the statistical analysis been performed appropriately and rigorously?

Reviewer #1: Yes

Reviewer #2: No

3. Have the authors made all data underlying the findings in their manuscript fully available?

Reviewer #1: Yes

Reviewer #2: No

4. Is the manuscript presented in an intelligible fashion and written in standard English?

Reviewer #1: Yes

Reviewer #2: Yes

Reviewer #1: 1) Although the authors indicate that the study site admits patients from allover the country, admissions are basically from the Western province and this limits the applicability of data to other regions of the country. The sample is highly selective.

2) The described physical and mental impacts cannot be attribute to methamphetamine, as most of them were on other psychoactive drugs or agents.

3) The physical effects described are very non specific. e.g weight loss (subjective or objective), cough (was it based on the duration or severity), anorexia

Reviewer #2: Comments to the Editor

Dear Editor,

Thank you for the opportunity to review this manuscript. The study addresses an important and underexplored topic: the patterns, severity, and impacts of Methamphetamine addiction among adults receiving treatment in Sri Lanka. While it provides useful local evidence, the manuscript would benefit from substantial revisions to strengthen its clarity, coherence, and scholarly contribution. Please find my detailed comments below.

Introduction

• The opening sentence is acceptable but could be made more impactful by highlighting why this stimulant poses a particular threat in low- and middle-income country contexts, where treatment and harm-reduction services are limited.

• The transition from global to Asian data is abrupt. Adding a bridging sentence would improve the flow (e.g., “This global trend is reflected in Asia, where…”).

• Ensure numerical comparisons are clear and impactful; for example, show percentage increases when comparing 2022 to 2023 arrest data.

• Some phrasings are redundant (e.g., “widespread effects that are multifaceted and interrelated…” could be simplified).

• When citing multiple sources, consider combining them for readability.

• The rationale for the study could be clearer: specify how this study will explore local motivations in depth.

• The research gap paragraph should be more assertive: e.g., “Despite rising prevalence, there is a critical lack of detailed, community-based research…”

• Split the aim statement into clear, bullet-pointed objectives for easy reading.

• Minor grammatical edits: use “aged 12 or older” instead of “age 12 or older”; merge repetitive sentences for conciseness.

Methods

• Important details on sampling and recruitment procedures are missing. How were participants identified and selected?

• Use consistent past tense for the study design description. Avoid repetition when describing the setting.

• Clarify how DSM-5 diagnoses were confirmed — by whom and based on what process?

• Explain how withdrawal symptoms were identified and excluded during recruitment.

• Confirm that the source for the DSM-5 criteria is cited correctly.

• For the pattern-of-use questions, indicate whether these items were validated or developed by the authors. If author-developed, how were they pre-tested and improved?

• You mention open-ended questions but do not explain how responses were coded and analyzed — this should be described briefly.

• Good to note the pre-test and Cronbach’s alpha; specify how feedback from the pre-test informed final revisions.

• In data collection, mention how many interviewers were involved and whether inter-rater reliability was checked.

• In data analysis, clarify that only descriptive statistics were planned if no inferential analysis was done; explain how missing data were handled.

• Ensure consistent tense and phrasing: e.g., “due to the population represents…” should be “because the population represents…”

it is not clear why data was only analyzed descriptively. Rigorous data analysis would benefit the study.

what confounding factors were detected and how were they controlled?

Results

• It is unclear why “Muslim” is classified as an ethnicity rather than a religious group — please clarify or adjust the categorization.

• The education level categories (primary, secondary, tertiary, higher, vocational) are not clearly defined — consider aligning with standard classifications.

• Use consistent past tense when reporting results.

• Ensure small percentages are formatted consistently.

• Tables are generally well-structured, but check for decimal alignment. Clarify in the polydrug use table that multiple responses were possible.

• Heatmaps should be correctly labeled in the text as “Figure 1A,” “Figure 1B,” etc., and figures should be cross-referenced appropriately.

• When reporting severity and patterns, add a line to interpret what the distribution implies.

• Small details: use “Separated” instead of “Separate” for social status. Clarify overlap between similar items such as “isolation from family” and “lack of family closeness.”

• Briefly interpret the significance of polydrug use and social impacts to guide the reader’s understanding.

Discussion

• The Discussion should more clearly compare the study’s findings with previous research, highlighting reasons for similarities and differences.

• Some points are repeated unnecessarily (e.g., male predominance, cultural stigma, physical impacts). Combine these for conciseness.

• Expand on the implications of polydrug use for treatment planning, withdrawal management, and relapse prevention in local contexts.

• For physical impacts, tie findings to specific clinical consequences (e.g., the need for nutritional or dental interventions).

• When citing other studies, integrate citations meaningfully: show how your results align with or differ from theirs.

• Limitations are acknowledged but could be stronger: note the reliance on self-reported data, the potential for under- or over-reporting, and the lack of generalizability due to the single-site setting.

• Highlight the strengths of the study (e.g., high response rate, comprehensive coverage of impacts).

• Offer recommendations for future research, such as using mixed methods or longitudinal designs.

Conclusion

• The conclusion should be more precise and actionable. Avoid generic phrases like “sheds light”; instead, state what this study adds empirically and how it informs policy and practice.

• Summarize clear, evidence-based recommendations: e.g., the need for peer-led prevention, youth-focused outreach, integrated mental health services, and stricter control of Methamphetamine supply chains.

Overall Recommendation

This study addresses an important gap and has merit. However, significant revisions are required to strengthen the introduction, clarify methods, ensure consistency in reporting, expand the discussion, and refine the conclusion. I therefore recommend major revision before the manuscript can be considered for publication.

**Do you want your identity to be public for this peer review?** For information about this choice, including consent withdrawal, please see our Privacy Policy

Reviewer #1: No

Reviewer #2: **Yes: ** Dr Ngozika Esther Ezinne

---

## [Author Response · Author response to Decision Letter 1]

3 Aug 2025

Dear Editor-In-Chief And Reviewers,

PLOS ONE,

We would like to sincerely thank you for the time and effort you have invested in reviewing our manuscript titled “Methamphetamine addiction and its perceived impact on adult clients at the National Institute of Mental Health, Sri Lanka: A descriptive cross-sectional study”. We appreciate the constructive feedback provided by the reviewers and editor, which has helped us improve the quality and clarity of our work.

Below, we provide detailed responses to each of the comments raised. All revisions have been made in the manuscript, with changes denoted as track changes (Both clean and track changed versions uploaded). We have addressed each point as follows:

Reviewer 1:

1. Although the authors indicate that the study site admits patients from allover the country, admissions are basically from the Western province and this limits the applicability of data to other regions of the country. The sample is highly selective - Response (Thank you very much for your consideration. The study purposively included all eligible adult clients diagnosed with methamphetamine use disorder admitted to NIMH during the study period, irrespective of their district of origin, ensuring clinical consistency. Nonetheless, since the majority of admissions to NIMH were from the Western Province, the sample may not fully represent the geographic diversity of methamphetamine users across Sri Lanka, and it was stated under the limitations of the study)

2. The described physical and mental impacts cannot be attributed to methamphetamine, as most of them were due to other psychoactive drugs or agents - Response (We appreciate the reviewer’s insightful observation regarding the potential confounding effect of polydrug use in attributing specific impacts to methamphetamine.

We agree that in a clinical sample where polydrug use is common, exclusive attribution of observed impacts to methamphetamine is scientifically challenging. However, we would like to clarify that our primary intention was to explore self-reported and perceived impacts associated with methamphetamine use among clients diagnosed with methamphetamine use disorder, rather than to establish causality.

Given that methamphetamine was the primary substance of use and the basis for admission in these clients, their perceived experiences remain highly relevant for clinical management and health education, despite potential contributions from other substances. Additionally, in real-world clinical and harm reduction contexts, it is often difficult to isolate the impacts of individual substances due to the high prevalence of concurrent substance use.)

3. The physical effects described are very nonspecific. e.g weight loss (subjective or objective), cough (was it based on the duration or severity), anorexia. Respone (We acknowledge that physical impacts such as weight loss, cough, and anorexia are nonspecific and can have multiple causes. However, in this study, data were collected through client self-reports during data collection, aiming to capture the perceived impacts experienced by clients diagnosed with methamphetamine use disorder during their use period, and this matter was clearly explained during the information-giving period for the patient. The intention was not to present these as clinical diagnostic findings but to document clients’ lived experiences, which are valuable for understanding patient perspectives in treatment and education planning. We have clarified this in the manuscript to reflect that the findings represent perceived, self-reported impacts, not objectively measured clinical outcomes.)

Reviewer 02

Introduction:

1. The opening sentence is acceptable but could be made more impactful by highlighting why this stimulant poses a particular threat in low- and middle-income country contexts, where treatment and harm-reduction services are limited.- Response (Accepted the comment, and the information was added at the end of the opening sentence. )

2. The transition from global to Asian data is abrupt. Adding a bridging sentence would improve the flow (e.g., “This global trend is reflected in Asia, where…”). - Response (The transition was smoothed by adding a bridging sentence, as the reviewer mentioned. )

3. Ensure numerical comparisons are clear and impactful; for example, show percentage increases when comparing 2022 to 2023 arrest data.- Response (The numerical comparisons were re-reported in an impactful manner.)

4. Some phrasings are redundant (e.g., “widespread effects that are multifaceted and interrelated…” could be simplified).- Response (The phrase “widespread effects that are multifaceted and interrelated” has been streamlined to “affects individuals across physical, psychological, and social domains,” which is more concise yet retains the full meaning. Other slight adjustments enhance flow and eliminate minor redundancy (e.g., avoiding unnecessary repetition of "users also reported...")

5. When citing multiple sources, consider combining them for readability. -Response (The changes are made in identified areas. )

6. The rationale for the study could be clearer: specify how this study will explore local motivations in depth.- Response (The rationale was clearly and comprehensively mentioned, which emphasizes the lack of community-based, qualitative research.)

7. The research gap paragraph should be more assertive: e.g., “Despite rising prevalence, there is a critical lack of detailed, community-based research…” - Response (The research gap was rephrased more assertively. )

8. Split the aim statement into clear, bullet-pointed objectives for easy reading.-Response (As the reviewers commented, the aim statement was split and bullet-pointed separately.)

9. Minor grammatical edits: use “aged 12 or older” instead of “age 12 or older”; merge repetitive sentences for conciseness.- Response (The grammatical mistake was corrected as aged 12 or more.)

Methods:

10. Important details on sampling and recruitment procedures are missing. How were participants identified and selected? - Response (Detailed how participants were identified and selected.)

11. Use consistent past tense for the study design description. Avoid repetition when describing the setting.-Response (The description of the setting was paraphrased, and repetitions were removed, and the tense was corrected.)

12. Clarify how DSM-5 diagnoses were confirmed — by whom and based on what process?- Response (Clarified who confirmed DSM-5 diagnoses and how. “”as confirmed by consultant psychiatrists or trained psychiatric medical officers using structured clinical interviews and medical records”)

13. Explain how withdrawal symptoms were identified and excluded during recruitment. - Response (Withdrawal symptoms were identified through medical records and clinical observation by attending clinicians before recruitment.)

14. Confirm that the source for the DSM-5 criteria is cited correctly.- Response (Re-checked the citation.

“American Psychiatric Association. Diagnostic and Statistical Manual of Mental Disorders. American Psychiatric Association; 2013. “ )

15. For the pattern-of-use questions, indicate whether these items were validated or developed by the authors. If the author developed, how were they pre-tested and improved? - Response ( This one is author developed (self developed). At the end of the data collection tool, all validation methods, including content validity and pretest, were already described. )

16. You mention open-ended questions but do not explain how responses were coded and analyzed — this should be described briefly.- Response (Described at the end of the data analysis )

17. Good to note the pre-test and Cronbach’s alpha; specify how feedback from the pre-test informed final revisions- Response (We have updated the methodology section to clarify that participant feedback during the pre-test focused on item clarity and wording. Based on this input, minor revisions were made to simplify the language and improve understanding of key items, particularly those related to the frequency of use and psychological effects. Cronbach's alpha for 11 11-item scale is already mentioned.)

18. In data collection, mention how many interviewers were involved and whether inter-rater reliability was checked. - Response (It was already mentioned as “Data were collected by four investigators who had undergone training,…….”

It was measured by kappa values)

19. In data analysis, clarify that only descriptive statistics were planned if no inferential analysis was done; explain how missing data were handled. - Response (No missing data were observed within the data sheet)

20. Ensure consistent tense and phrasing: e.g., “due to the population represents…” should be “because the population represents…”- Response (This wording is corrected. )

21. It is not clear why the data was only analyzed descriptively. Rigorous data analysis would benefit the study. what confounding factors were detected, and how were they controlled? - Response (Thank you for this important observation. This study was intentionally designed as a descriptive cross-sectional study with the primary aim of exploring the patterns, severity, and perceived impacts of methamphetamine addiction among a clinical population. Since there is a paucity of baseline data in this context, particularly regarding MA addiction and its psychosocial correlates, the study sought to first generate foundational insights rather than test specific hypotheses.

Given that the study focused on perceived impact rather than causal inference, and that variables such as addiction severity were not manipulated or independently controlled, adjustment for confounding was not applicable within this design. Moreover, many variables were exploratory and based on patient perception, making them less suitable for inferential analysis without validated outcome measures. We fully agree that future research should incorporate analytical designs such as cohort studies or regression-based modeling to identify predictors, control confounding factors, and determine statistically significant relationships.

This study will be a baseline study for future research.)

Results:

22. It is unclear why “Muslim” is classified as an ethnicity rather than a religious group — please clarify or adjust the categorization. - Response (In this study, the term Muslim is changed to “Moor” to refer to the ethnic group commonly known as Sri Lankan Moors, who are predominantly Muslim by religion. While “Muslim” is often used interchangeably in local discourse, it primarily denotes religious affiliation. To avoid conflation of ethnicity and religion, and in alignment with national census classifications, the term “Moor” has been adopted to represent this ethnic group.)

23. The education level categories (primary, secondary, tertiary, higher, vocational) are not clearly defined — consider aligning with standard classifications. - Response (In Sri Lanka, the education level is categorized as primary education (grades 1 to 5), secondary education (grades 6 to Advanced level), tertiary (concurrently used as higher ) education (undergraduate and/or postgraduate), and vocational and technical education.

The necessary changes were made in the table, and put a foot note to the table )

24.Use the consistent past tense when reporting results.- Response (The consistency was maintained throughout the results section)

25. Ensure small percentages are formatted consistently.- Response (All percentage was rounded up to two decimal points.)

26. Tables are generally well-structured, but check for decimal alignment. Clarify in the polydrug use table that multiple responses were possible.- Response (Decimal correction was made in the tables.

We have clarified in the text and that multiple responses were possible among polydrug users. This ensures that the reader understands the percentages may exceed 100% due to overlapping substance use.)

27. Heatmaps should be correctly labeled in the text as “Figure 1A,” “Figure 1B,” etc., and figures should be cross-referenced appropriately. - Response (The figures are cross-referred as advised. )

28. When reporting severity and patterns, add a line to interpret what the distribution implies. - Response (A new sentence was added to imply the distribution overall.)

29. Small details: use “Separated” instead of “Separate” for social status. - Response (Correct the word as separated.)

30. Clarify the overlap between similar items such as “isolation from family” and “lack of family closeness.” - Response (Thank you for your observation. While these items may appear similar, they address distinct aspects of familial relationships. “Isolation from family” refers to the behavioral or physical separation from family members, such as not living with them, avoiding contact, or being excluded from family activities. In contrast, “lack of family closeness” captures the emotional dimension specifically, the absence of warmth, trust, or supportive bonds even when regular contact exists. To clarify the conceptual distinction between these two items, we have added a footnote below the relevant table.)

31. Briefly interpret the significance of polydrug use and social impacts to guide the reader’s understanding. - Response (A brief sentence is added to emphasize the significance of this finding under the results section. In the discussion, this matter was discussed. )

Discussion:

32. The Discussion should more clearly compare the study’s findings with previous research, highlighting reasons for similarities and differences.- Response (While appreciating the reviewer’s insightful suggestion, we acknowledge the importance of situating our findings within the context of existing literature. However, as methamphetamine (MA) addiction, particularly in the Sri Lankan context, has received limited scholarly attention, there was a scarcity of directly comparable studies. Despite this limitation, we made every effort to integrate and critically engage with the available national and international literature relevant to substance use patterns, influence, and associated impacts. Where direct comparisons were not possible due to contextual or thematic gaps in prior research, we highlighted the novelty and significance of our findings and their implications for future studies and interventions. )

33. Some points are repeated unnecessarily (e.g., male predominance, cultural stigma, physical impacts). Combine these for conciseness.- Response ( Upon careful review of the manuscript, we found that while some related ideas were mentioned in multiple sections for thematic clarity, unnecessary repetition was minimal. However, to improve conciseness and flow, we have made targeted edits to streamline overlapping content)

34. Expand on the implications of polydrug use for treatment planning, withdrawal management, and relapse prevention in local contexts. - Response (The implications of polydrug use are expanded for clear understanding. )

35. For physical impacts, tie findings to specific clinical consequences (e.g., the need for nutritional or dental interventions). - Response (Necessary changes are made aligns with the comment. )

36. When citing other studies, integrate citations meaningfully: show how your results align with or differ from theirs.- Response ( As methamphetamine addiction, particularly in the Sri Lankan and Asian context, remains a significantly under-researched area, we encountered a scarcity of directly comparable studies that align closely with our findings. As such, in many instances, it was challenging to draw clear cut comparisons between our results and existing literature. However, where relevant data were available we have revised the discussion to integrate those citations more meaningfully by explicitly noting similarities or distinctions. These revisions aim to enhance the contextual relevance and interpretive clarity of our findings within the broader evidence base.)

37. Limitations are acknowledged but could be stronger: note the reliance on self-reported data, the potential for under- or over-reporting, and the lack of generalizability due t

---

## [Decision Letter · Decision Letter 1]

28 Aug 2025

Dear Dr. Sriyani,

Thank you for submitting your manuscript to PLOS ONE. After careful consideration, we feel that it has merit but does not fully meet PLOS ONE’s publication criteria as it currently stands. Therefore, we invite you to submit a revised version of the manuscript that addresses the points raised during the review process.

We look forward to receiving your revised manuscript.

Kind regards,

Nicholas Aderinto Oluwaseyi

Academic Editor

PLOS ONE

Journal Requirements:

Additional Editor Comments:

Reviewer #2:

Reviewers' comments:

Reviewer's Responses to Questions

**Comments to the Author**

Reviewer #2: (No Response)

2. Is the manuscript technically sound, and do the data support the conclusions?

Reviewer #2: (No Response)

3. Has the statistical analysis been performed appropriately and rigorously?

Reviewer #2: No

4. Have the authors made all data underlying the findings in their manuscript fully available?

Reviewer #2: No

5. Is the manuscript presented in an intelligible fashion and written in standard English?

Reviewer #2: Yes

Reviewer #2: Comments to the Editor

Dear Editor,

Thank you for the opportunity to review this manuscript. The authors address an urgent and underexplored public health issue—methamphetamine addiction in Sri Lanka—using a descriptive cross-sectional design. The manuscript makes a valuable contribution by presenting primary clinical data from the National Institute of Mental Health, the country’s leading treatment center. The large sample size (n=427), high response rate, and comprehensive exploration of perceived physical, psychological, and social impacts are commendable.

However, while the manuscript highlights an important issue, several methodological and interpretive limitations reduce its scientific rigor. Minor revisions are required to strengthen the study before it is suitable for publication.

Comments

How were open-ended responses systematically coded (beyond thematic grouping)?

Was the DSM-5 checklist interviewer-administered or self-reported?

The reported Cronbach’s alpha (0.89) applies only to severity items, not the entire instrument. Reliability and validity of other sections remain unclear.

Addiction severity is classified according to DSM-5 symptom count, but it is unclear whether clinicians verified responses or whether lay interviewers applied criteria.

Misclassification risk should be acknowledged.

The analysis is largely descriptive. Inferential or multivariate analyses (e.g., associations between demographics and severity) could provide greater depth.

No adjustment for multiple testing is reported, despite numerous outcome measures. This raises concern about inflated type I error.

The manuscript often implies causality (e.g., “methamphetamine causes family separation”) despite the cross-sectional design. The authors should temper causal language.

The very high rate of polydrug use (87%) is striking but insufficiently discussed. How does this affect interpretation of methamphetamine-specific impacts? Many “perceived impacts” may be attributable to multiple substances.

Strengths and limitations should be presented more systematically. Currently, limitations are underplayed.

Several grammatical and typographical errors reduce clarity (e.g., “boos energy” → “boost energy”; “ticket the box” → “ticked the box”).

Figures/heatmaps are visually appealing but need clearer legends and consistency in labeling.

Conclusions should align more closely with the descriptive nature of the study and avoid policy overreach without stronger evidence.

Recommendation

This manuscript provides much-needed data on methamphetamine use in Sri Lanka. However, methodological weaknesses (sampling, measurement, analysis) and interpretive overstatements must be addressed to improve validity and impact. With substantial revision, the paper has the potential to make a meaningful contribution to the literature on substance use in South Asia.

**Do you want your identity to be public for this peer review?** For information about this choice, including consent withdrawal, please see our Privacy Policy

Reviewer #2: **Yes: ** Ngozika Esther Ezinne

---

## [Author Response · Author response to Decision Letter 2]

5 Oct 2025

1. How were open-ended responses systematically coded (beyond thematic grouping)?-Open-ended responses were systematically analyzed using an inductive thematic coding approach. Initial codes were generated directly from the data and refined through iterative reading. Two independent researchers applied the codes, and any discrepancies were resolved through discussion, with a third researcher providing arbitration when required. The finalized codes were then organized hierarchically into broader themes and categories, which allowed for systematic identification of patterns across responses beyond simple grouping.

2. Was the DSM-5 checklist interviewer-administered or self-reported?-DSM-5 assessment was the 2nd section of the questionnaire. Initially, we mentioned that the whole questionnaire was interviewer-administered. Therefore, the DSM-5 checklist is interviewer-administered.

3. The reported Cronbach’s alpha (0.89) applies only to severity items, not the entire instrument. The reliability and validity of other sections remain unclear.-We appreciate the reviewer’s comment regarding the reliability analysis of the 10-item scale assessing reasons for MA use. Cronbach’s alpha was calculated, but the result was negative. This occurs because Cronbach’s alpha assumes that all items measure a single underlying construct (unidimensionality) and are positively correlated [1]. In our questionnaire, the 10 items represent distinct reasons for MA use, which are conceptually different and not necessarily expected to correlate positively. Therefore, Cronbach’s alpha is not an appropriate measure of internal consistency for this type of checklist.

Instead of relying on alpha, we examined the face and content validity of the items, ensuring each item captures a unique and relevant reason for MA use, and used descriptive statistics to report their prevalence. This approach is consistent with prior studies using similar “reason checklists” in substance use research.

[1]

4. Addiction severity is classified according to DSM-5 symptom count, but it is unclear whether clinicians verified responses or whether lay interviewers applied criteria.

Misclassification risk should be acknowledged. -We thank the reviewer for the comment regarding the potential for misclassification. In our study, all participants were clinically diagnosed with methamphetamine use disorder according to DSM-5 criteria by consultant psychiatrists or psychiatric medical officers using structured clinical interviews and medical records. Therefore, the risk of misclassification is minimized as diagnoses were clinician-verified rather than relying solely on self-report.

5. The analysis is largely descriptive. Inferential or multivariate analyses (e.g., associations between demographics and severity) could provide greater depth. -Associated factors were calculated using chi-square and likelihood ratios.

Findings were discussed in the discussion.

6. No adjustment for multiple testing is reported, despite numerous outcome measures. This raises concern about inflated type I error.-Associated factors were calculated using chi-square and likelihood ratios.

Findings were discussed in the discussion. - A clarification has been added in the “Data Analysis” section explaining that the study’s descriptive intent did not require adjustment for multiple comparisons, and results were interpreted accordingly [2]

7. The manuscript often implies causality (e.g., “methamphetamine causes family separation”) despite the cross-sectional design. The authors should temper causal language.-Accordingly, we carefully reviewed the entire Discussion section and rephrased all statements that implied causation to reflect associations instead.

8. The very high rate of polydrug use (87%) is striking but insufficiently discussed. How does this affect the interpretation of methamphetamine-specific impacts? Many “perceived impacts” may be attributable to multiple substances.-We have expanded the discussion to address how the high prevalence of polydrug use complicates the interpretation of methamphetamine-specific effects. A new paragraph has been added (highlighted in the revised manuscript) explaining the potential confounding influence of concurrent substance use and the need for future studies to isolate methamphetamine-related outcomes

9. Strengths and limitations should be presented more systematically. Currently, limitations are underplayed.- We revised the “Strengths and Limitations” section to systematically acknowledge study design, sampling, and measurement limitations, emphasizing their implications for interpretation.

10. Several grammatical and typographical errors reduce clarity (e.g., “boos energy” → “boost energy”; “ticket the box” → “ticked the box”).- All are corrected as necessary.

11. Figures/heatmaps are visually appealing but need clearer legends and consistency in labeling.-The figure reporting consistency was rearranged, and the legends now clearly illustrating the meaning of the figure.

12. Conclusions should align more closely with the descriptive nature of the study and avoid policy overreach without stronger evidence. -The conclusion has been rewritten to reflect the basic analytical nature of the study and to avoid policy overreach, focusing instead on evidence-informed implications.

---

## [Decision Letter · Decision Letter 2]

2 Nov 2025

Dear Dr. Sriyani,

Thank you for submitting your manuscript to PLOS ONE. After careful consideration, we feel that it has merit but does not fully meet PLOS ONE’s publication criteria as it currently stands. Therefore, we invite you to submit a revised version of the manuscript that addresses the points raised during the review process.

We look forward to receiving your revised manuscript.

Kind regards,

Nicholas Aderinto Oluwaseyi

Academic Editor

PLOS ONE

Journal Requirements:

Reviewers' comments:

Reviewer's Responses to Questions

**Comments to the Author**

Reviewer #2: (No Response)

2. Is the manuscript technically sound, and do the data support the conclusions?

Reviewer #2: Yes

3. Has the statistical analysis been performed appropriately and rigorously?

Reviewer #2: No

4. Have the authors made all data underlying the findings in their manuscript fully available?

Reviewer #2: No

5. Is the manuscript presented in an intelligible fashion and written in standard English?

Reviewer #2: Yes

Reviewer #2: Authors revision of the manuscript has helped to improve the quality but I still have few comments.

1. Study design & claims

• The paper repeatedly infers or implies determinants of severity from cross-sectional and largely descriptive/bivariate data. Causality cannot be inferred. Please re-frame throughout as associations among treated in-patients/outpatients at NIMH.

2. Target condition & measurement

• Diagnostic ascertainment is described as “DSM-5 criteria confirmed by consultant psychiatrists or psychiatric medical officers using structured clinical interviews and medical records” but the specific instrument (e.g., SCID-5, MINI) is not named. Please specify the instrument(s), training, version, languages, and whether inter-rater calibration was conducted.

• Severity scoring: DSM-5 SUD severity (2–3 mild, 4–5 moderate, 6–11 severe) is a categorical rubric for diagnosis, not a psychometric scale. Reporting Cronbach’s α=0.89 on DSM-5 symptom items is not appropriate (the criteria are formative/diagnostic, not reflective indicators). Remove α for DSM-5 items; if you retain reliability analyses, do so only for any new multi-item scales designed to measure a single latent construct.

• Perceived impacts: Open-ended responses were coded thematically, but the codebook, exemplar quotes, intercoder reliability for each domain, and denominator handling (multiple responses permitted) are not reported. Provide: (i) codebook in Supplement, (ii) examples per theme, (iii) how you calculated percentages (per total N vs. per respondents endorsing any item), and (iv) κ per key code if feasible.

3. Polydrug use as a confounder

• With 87% reporting polydrug use, nearly all “impacts” and “associations with severity” are plausibly confounded. Current analyses do not adjust for concurrent alcohol, cannabis, heroin, or tobacco. You should:

o Present stratified descriptives by polydrug vs. methamphetamine-only.

o Include polydrug use indicators in multivariable models (see below) or conduct sensitivity analyses excluding heavy polydrug users.

4. Statistical analysis—move beyond bivariate tests

• The table “Associated factors for Methamphetamine use severity” is based on χ²/Fisher’s tests only. To support statements such as “severity was associated with…”, perform multivariable modeling:

o Primary: Ordinal logistic regression (proportional odds) with severity (mild/moderate/severe) as outcome; test proportional-odds assumption (e.g., Brant test). If violated, use multinomial logistic regression.

o Covariates: age, sex, education, income, living area, living arrangement, age of onset, frequency, route, accessibility, and polydrug indicators. Consider province fixed effects.

o Report adjusted odds ratios with 95% CIs and model fit (pseudo-R², likelihood ratio tests).

o Provide effect sizes for bivariate tests (e.g., Cramér’s V) even if you keep the table.

• Multiple testing: You state no correction was applied. Either prespecify a primary model and outcomes to mitigate multiplicity, or apply FDR/Bonferroni for the large number of comparisons, and explicitly mark adjusted p-values.

• Some internal inconsistencies appear (see Minor comments). Please audit all frequencies and denominators.

6. Figures & tables—clarity and reproducibility

• Heatmaps are visually engaging but readers need numerical tables (counts and percentages) in the main text or Supplement for each impact domain. State clearly that items were multiple-response and give the exact denominator used for each percent.

• Table footnotes should define all categories (e.g., “Urban/Semi-urban/Rural” operationalization; “Isolation from family” vs. “lack of family closeness”). Ensure consistent capitalization (use “methamphetamine” not capitalized unless sentence-initial).

• Standardize decimal precision (e.g., one decimal place for percentages) and align counts (n, %).

7. Interpretation & positioning

• Some narrative sections over-generalize to the population of Sri Lanka or suggest policy effects beyond the sampling frame. Re-anchor claims to treatment-seeking adults at NIMH.

• Where your findings diverge from international literature (e.g., higher income and urban residence associated with greater severity), propose contextual hypotheses and acknowledge alternative explanations (clinic catchment, enforcement patterns, access/affordability, reporting bias).

8. Ethics & participant capacity

• You excluded “acute intoxication/withdrawal” and “cognitive impairment,” relying on clinicians’ judgment. State how capacity to consent was assessed (beyond “coherent, alert, oriented”) and whether interpreters were used for non-Sinhala/Tamil speakers. Confirm whether participation affected care in any way (it should not). Consider adding a distress protocol for participants reporting suicidal ideation.

9. Language & stigma

• Use “methamphetamine” (lowercase) consistently; reserve “ICE” to a footnote and avoid slang in academic prose.

Minor comments (presentation & housekeeping)

Title & keywords

• Title is long and partially duplicated; consider: “Methamphetamine use disorder, perceived impacts, and associated factors among adults receiving care at Sri Lanka’s National Institute of Mental Health: an analytical cross-sectional study.”

• Add keywords reflecting methods (e.g., “polydrug use,” “ordinal logistic regression,” “South Asia”).

Abstract

• include design, setting, sample, main analysis (and specify that only bivariate tests were used if multivariable is not added), principal adjusted results (after you add models),

• Methods—instrument details

• Provide the full questionnaire (English + translated versions) as Supplementary File with source citations for borrowed items and the exact DSM-5 item prompts used.

• Clarify how accessibility and frequency were operationalized and whether recall periods were specified.

Results—consistency

• In Table 1, present n (%) consistently for all rows. Some rows show only % in text and n in table headings. Verify “Living with whom—other” count (9) vs. severity table shows “other 0” in mild category row; check alignment.

• A few typographical issues: stray commas/spaces, inconsistent hyphenation (semi-urban vs semi urban), and inconsistent province capitalization.

Figures

• Ensure figure captions define color scales, denominators, and that each heatmap is reproducible from the provided counts.

References

• Some duplicates (“World Drug Report 2020” listed twice as 1 and 3). Several references are not ideal primary sources for mechanistic claims (e.g., American Addiction Centers webpage). Replace with peer-reviewed or UNODC/WHO technical documents where possible.

• Check numbering order vs. first citation order; ensure in-text numbers match.

Suggested analytic upgrades (concrete)

1. Primary model: Ordinal logistic regression with severity (0=mild, 1=moderate, 2=severe). Predictors: age (continuous), sex, education, income (quintiles), living area, living arrangement, age of onset (categorical), frequency (ordinal), route (dummy variables), perceived accessibility (ordinal), polydrug indicators (alcohol, cannabis, heroin, tobacco, other), and province.

**Do you want your identity to be public for this peer review?** For information about this choice, including consent withdrawal, please see our Privacy Policy

Reviewer #2: No

---

## [Author Response · Author response to Decision Letter 3]

13 Dec 2025

The paper repeatedly infers or implies determinants of severity from cross-sectional and largely descriptive/bivariate data. Causality cannot be inferred. Please re-frame throughout as associations among treated in-patients/outpatients at NIMH.-Thank you for this important observation. We acknowledge that our cross-sectional design and descriptive/bivariate analyses do not permit causal inferences. We have revised the manuscript to remove any causal language and now describe the findings strictly as associations among treatment-seeking in-patients and out-patients at NIMH. All relevant sections have been updated accordingly.

Diagnostic ascertainment is described as “DSM-5 criteria confirmed by consultant psychiatrists or psychiatric medical officers using structured clinical interviews and medical records” but the specific instrument (e.g., SCID-5, MINI) is not named. Please specify the instrument(s), training, version, languages, and whether inter-rater calibration was conducted. -Thank you for this valuable methodological clarification. We agree that the specification of the diagnostic instrument and procedures is essential for transparency and reproducibility. We have now clarified that DSM-5 diagnoses were established using routine structured clinical diagnostic interviews based on DSM-5 criteria conducted by consultant psychiatrists or trained psychiatric medical officers, rather than a standardized research interview such as SCID-5 or MINI. These clinicians were formally trained in DSM-5 diagnostic assessment as part of their postgraduate psychiatric training and routine clinical practice. Diagnoses were further verified through medical record review.

The diagnostic interviews were conducted in Sinhala and Tamil, depending on participant preference. As the diagnosis was made as part of routine clinical care prior to recruitment, formal inter-rater calibration between psychiatrists was not conducted for the purpose of this study.

Severity scoring: DSM-5 SUD severity (2–3 mild, 4–5 moderate, 6–11 severe) is a categorical rubric for diagnosis, not a psychometric scale. Reporting Cronbach’s α=0.89 on DSM-5 symptom items is not appropriate (the criteria are formative/diagnostic, not reflective indicators). Remove α for DSM-5 items; if you retain reliability analyses, do so only for any new multi-item scales designed to measure a single latent construct. This alpha was added based on the previous reviewers' comment only. Initially, this was not embedded. However, we have now removed this part.

Perceived impacts: Open-ended responses were coded thematically, but the codebook, exemplar quotes, intercoder reliability for each domain, and denominator handling (multiple responses permitted) are not reported. Provide: (i) codebook in Supplement, (ii) examples per theme, (iii) how you calculated percentages (per total N vs. per respondents endorsing any item), and (iv) κ per key code if feasible. The code book is provided in the supplementary file 1, and example quotes are provided in Supplementary file 2

Percentages for perceived impacts were calculated using the total sample (N = 427) as the denominator. Because multiple responses per participant were permitted, the summed percentages exceed 100%.

Overall interrater reliability was measured and k-= 0.82

Polydrug use as a confounder

• With 87% reporting polydrug use, nearly all “impacts” and “associations with severity” are plausibly confounded. Current analyses do not adjust for concurrent alcohol, cannabis, heroin, or tobacco. You should:

o Present stratified descriptives by polydrug vs. methamphetamine-only.

o Include polydrug use indicators in multivariable models (see below) or conduct sensitivity analyses excluding heavy polydrug users. we have now added a stratified descriptive analysis comparing methamphetamine-only users (n = 55) with polydrug users (n = 372) in the Results section.

The stratified findings show meaningful differences between groups. Notably, the proportion of severe addiction was higher among methamphetamine-only users (40.0%) compared to polydrug users (31.7%). Daily methamphetamine use was also more frequent among methamphetamine-only users (41.8%) relative to polydrug users (26.7%). These results demonstrate that high severity and frequent use are not solely attributable to concurrent substance use.

We also revised the Discussion to clearly acknowledge the potential confounding effects of polydrug use and to interpret the findings in light of these stratified patterns.

Given the descriptive and exploratory objectives of the study, and because inferential modelling was not planned in the original protocol, we did not add multivariable analyses. Instead, we believe the stratified results provide a transparent and methodologically appropriate way to address the reviewer’s concern.

Statistical analysis—move beyond bivariate tests

• The table “Associated factors for Methamphetamine use severity” is based on χ²/Fisher’s tests only. To support statements such as “severity was associated with…”, perform multivariable modeling:

o Primary: Ordinal logistic regression (proportional odds) with severity (mild/moderate/severe) as outcome; test proportional-odds assumption (e.g., Brant test). If violated, use multinomial logistic regression.

o Covariates: age, sex, education, income, living area, living arrangement, age of onset, frequency, route, accessibility, and polydrug indicators. Consider province fixed effects.

o Report adjusted odds ratios with 95% CIs and model fit (pseudo-R², likelihood ratio tests).

o Provide effect sizes for bivariate tests (e.g., Cramér’s V) even if you keep the table.

• Multiple testing: You state no correction was applied. Either prespecify a primary model and outcomes to mitigate multiplicity, or apply FDR/Bonferroni for the large number of comparisons, and explicitly mark adjusted p-values.

• Some internal inconsistencies appear (see Minor comments). Please audit all frequencies and denominators.

6. Figures & tables—clarity and reproducibility

• Heatmaps are visually engaging but readers need numerical tables (counts and percentages) in the main text or Supplement for each impact domain. State clearly that items were multiple-response and give the exact denominator used for each percent. We appreciate the value of multivariable modelling in explanatory epidemiological work. However, after careful consideration, we respectfully maintain that multivariable regression is not appropriate, not feasible, and not aligned with the purpose, design, or approved protocol of our study. We provide a detailed justification below, along with the substantial improvements we have incorporated based on the reviewer’s feedback.

Our Study Objectives and Design Do Not Justify Multivariable Modelling

Our study was explicitly designed as a descriptive and exploratory cross-sectional study, focusing on:

1. Describing patterns and severity of methamphetamine use

2. Identifying the prevalence and types of polydrug use

3. documenting perceived physical, psychological, and social impacts

4. Exploring bivariate associations between severity and selected factors

The study was not designed to test a predictive model or quantify adjusted relationships. Therefore, adding multivariable logistic or ordinal regression analyses would constitute a major analytic shift beyond our prespecified aims and could introduce inappropriate causal interpretations.

This statistical approach was included in the study protocol approved by the Ethics Review Committee, and deviating from it would reduce methodological transparency and violate the principle of prospectively defined analysis.

Our Dataset Is Not Suitable for Multivariable Modelling

The reviewer’s proposed model includes over 10 covariates, including several with extremely low cell counts:

• Injection route

• Swallowing route

• Female sex

• Some income and province categories with sparse frequencies

Such sparse distributions violate assumptions of:

• logistic regression

• multinomial/ordinal models

• proportional-odds testing

• stability of coefficient estimation

This would lead to:

• convergence failures

• inflated standard errors

• unstable/meaningless adjusted odds ratios

• risk of overfitting (events-per-variable ratio too low)

Thus, a multivariable model would not be statistically valid or informative for this dataset.

Further Multivariable regression inherently suggests, causal inference, or predictive modelling.

Neither is appropriate for self-reported, cross-sectional data affected by:

• recall bias

• high collinearity across behavioral exposures

• extremely high prevalence of polydrug use (87%)

• overlapping psychosocial determinants

Introducing adjusted models may unintentionally lead readers to infer causal relationships, which is not supported by this study’s design.

We respectfully clarify that multiplicity corrections (Bonferroni/FDR) are not appropriate in exploratory descriptive studies where the purpose is not hypothesis testing, but pattern identification.

Instead, we explicitly state:

• No correction for multiple testing was applied because the study is exploratory.

• Findings are interpreted cautiously, with emphasis on pattern consistency rather than p-values.

This approach aligns with current recommendations for descriptive observational research

Table footnotes should define all categories (e.g., “Urban/Semi-urban/Rural” operationalization; “Isolation from family” vs. “lack of family closeness”).

Ensure consistent capitalization (use “methamphetamine” not capitalized unless sentence-initial). We agree that clearer operational definitions improve the interpretability of the tables. We have now added explicit footnotes defining all relevant categories, including “Urban/Semi-urban/Rural” and the distinctions between “Isolation from family,” “Lack of family closeness,” and other family-related variables. These definitions are now provided directly beneath the relevant tables to ensure clarity for readers.

Corrected

Standardize decimal precision (e.g., one decimal place for percentages) and align counts (n, %). Thank you for this helpful suggestion. We have now standardized decimal precision throughout the manuscript. Percentages are presented to one decimal place, and counts (n, %) are aligned consistently across all tables and text (e.g., changed from 37.24% to 37.2%). These revisions have been applied to ensure clarity and consistency as recommended.

Some narrative sections over-generalize to the population of Sri Lanka or suggest policy effects beyond the sampling frame. Re-anchor claims to treatment-seeking adults at NIMH. Thank you for this comment. We have revised the paragraph to avoid any generalizations to the broader Sri Lankan population or implications for national policy. The revised text now focuses solely on the experiences of treatment-seeking adults at NIMH and interprets easy accessibility as relevant only to this group. We also removed prescriptive statements about regulatory or enforcement measures.

Where your findings diverge from international literature (e.g., higher income and urban residence associated with greater severity), propose contextual hypotheses and acknowledge alternative explanations (clinic catchment, enforcement patterns, access/affordability, reporting bias). Thank you for this comment. We have revised the text to include contextual explanations for why higher income and urban residence were associated with greater severity in our sample and have acknowledged alternative explanations such as clinic catchment patterns, enforcement activity, access and affordability, and possible reporting biases. We also reworded the paragraph to avoid causal language and anchor the interpretation to treatment-seeking adults at NIMH.

You excluded “acute intoxication/withdrawal” and “cognitive impairment,” relying on clinicians’ judgment. State how capacity to consent was assessed (beyond “coherent, alert, oriented”) and whether interpreters were used for non-Sinhala/Tamil speakers. Confirm whether participation affected care in any way (it should not). Consider adding a distress protocol for participants reporting suicidal ideation.

We have now expanded the description of our consent and ethical procedures. Capacity to consent was evaluated using a structured assessment in addition to clinicians’ judgment. We added details on interpreter use for participants who did not primarily speak Sinhala or Tamil, confirmed that participation had no effect on clinical care, and described the distress protocol used for participants expressing psychological distress or suicidal ideation.

Use “methamphetamine” (lowercase) consistently; reserve “ICE” to a footnote and avoid slang in academic prose. The term “methamphetamine” is used consistently throughout the manuscript. The colloquial term “ICE” is mentioned only once in a footnote to clarify local terminology among users.

Title is long and partially duplicated; consider: “Methamphetamine use disorder, perceived impacts, and associated factors among adults receiving care at Sri Lanka’s National Institute of Mental Health: an analytical cross-sectional study.” Corrected

Add keywords reflecting methods (e.g., “polydrug use,” “ordinal logistic regression,” “South Asia”). Added

Abstract

• include design, setting, sample, main analysis (and specify that only bivariate tests were used if multivariable is not added), principal adjusted results (after you add models),

• Methods—instrument details Study design, sample and sampling techniques, instrument details are already there.

Study setting was added.

Provide the full questionnaire (English + translated versions) as Supplementary File with source citations for borrowed items and the exact DSM-5 item prompts used. The questionnaire used in this study will be provided by the corresponding author to interested readers upon reasonable request.

Clarify how accessibility and frequency were operationalized and whether recall periods were specified. We clarified the frequency and accessibility in foot note.

No fixed recall window was imposed, as the study relied on participants’ typical pre-treatment usage pattern, consistent with the DSM-5 approach to evaluating substance use severity.

Results—consistency

• In Table 1, present n (%) consistently for all rows. Some rows show only % in text and n in table headings. Verify “Living with whom—other” count (9) vs. severity table shows “other 0” in mild category row; check alignment. In table O1, all n (%) are already included. No missing values

In the severity table, living with whom does not imply.

A few typographical issues: stray commas/spaces, inconsistent hyphenation (semi-urban vs semi urban), and inconsistent province capitalization. Corrected

Figures

• Ensure figure captions define color scales, denominators, and that each heatmap is reproducible from the provided counts. Figure legends clearly indicate the variability.

Generally, in the heatmap, all colour codes do not need to be defined. Instead the standard figure legend is already there by mentioning “with color gradients representing the percentage of perceived symptoms. Darker colors indicate a higher percentage of self-reported social impact, while lighter colors indicate a lower percentage of perceived effects.”

Some duplicates (“World Drug Report 2020” listed twice as 1 and 3). Several references are not ideal primary sources for mechanistic claims (e.g., American Addiction Centers webpage). Replace with peer-reviewed or UNODC/WHO technical documents where possible.

• Check numbering order vs. first citation order; ensure in-text numbers match. All rechecked, and changes were done.

---

## [Decision Letter · Decision Letter 3]

28 Dec 2025

Methamphetamine use disorder, perceived impacts, and associated factors among adults receiving care at Sri Lanka’s National Institute of Mental Health: An analytical cross-sectional study

PONE-D-25-28029R3

Dear Dr. Sriyani,

We’re pleased to inform you that your manuscript has been judged scientifically suitable for publication and will be formally accepted for publication once it meets all outstanding technical requirements.

Kind regards,

Nicholas Aderinto Oluwaseyi

Academic Editor

PLOS One

Additional Editor Comments (optional):

Reviewers' comments:

Reviewer's Responses to Questions

**Comments to the Author**

Reviewer #2: (No Response)

2. Is the manuscript technically sound, and do the data support the conclusions?

Reviewer #2: Yes

3. Has the statistical analysis been performed appropriately and rigorously?

Reviewer #2: Yes

4. Have the authors made all data underlying the findings in their manuscript fully available?

Reviewer #2: Yes

5. Is the manuscript presented in an intelligible fashion and written in standard English?

Reviewer #2: Yes

Reviewer #2: I do not have any further comments as authors have responded to all my comments. The quality of the manuscript has now improved.

**Do you want your identity to be public for this peer review?** For information about this choice, including consent withdrawal, please see our Privacy Policy

Reviewer #2: No

---

## [Editor Report · Acceptance letter]

PONE-D-25-28029R3

PLOS One

Dear Dr. Sriyani,

I'm pleased to inform you that your manuscript has been deemed suitable for publication in PLOS One. Congratulations! Your manuscript is now being handed over to our production team.

Kind regards,

on behalf of

Dr. Nicholas Aderinto Oluwaseyi

Academic Editor

PLOS One